# Visible light-exposed lignin facilitates cellulose solubilization by lytic polysaccharide monooxygenases

Eirik G. Kommedal [1], Camilla F. Angeltveit[1], Leesa J. Klau [2], Iván Ayuso-Fernández[1], Bjørnar Arstad[3], Simen G. Antonsen[1], Yngve Stenstrøm[1], Dag Ekeberg [1], Francisco Gírio[4], Florbela Carvalheiro[4], Svein J. Horn [1], Finn Lillelund Aachmann [2] & Vincent G. H. Eijsink [1] ✉

Lytic polysaccharide monooxygenases (LPMOs) catalyze oxidative cleavage of crystalline polysaccharides such as cellulose and are crucial for the conversion of plant biomass in Nature and in industrial applications. Sunlight promotes microbial conversion of plant litter; this effect has been attributed to photochemical degradation of lignin, a major redox-active component of secondary plant cell walls that limits enzyme access to the cell wall carbohydrates. Here, we show that exposing lignin to visible light facilitates cellulose solubilization by promoting formation of $H_2O_2$ that fuels LPMO catalysis. Light-driven $H_2O_2$ formation is accompanied by oxidation of ring-conjugated olefins in the lignin, while LPMO-catalyzed oxidation of phenolic hydroxyls leads to the required priming reduction of the enzyme. The discovery that light-driven abiotic reactions in Nature can fuel $H_2O_2$-dependent redox enzymes involved in deconstructing lignocellulose may offer opportunities for bioprocessing and provides an enzymatic explanation for the known effect of visible light on biomass conversion.

Every year, 100 billion tons of $CO_2$ are converted to cellulose by photosynthetic organisms[1], making lignocellulosic plant biomass the most abundant natural material on Earth and a large reservoir of renewable carbon that can be transformed to chemicals and fuels. However, plant cell walls have evolved to become recalcitrant co-polymeric structures to provide mechanical strength and rigidity and to provide resistance against pathogen attack, and are, thus, hard to break down[2]. Plant cell wall-degrading microorganisms have solved this challenge by developing multi-component enzymatic tools that act synergistically to process this highly complex and recalcitrant biomass.

Selective oxidation of non-activated C-H bonds in crystalline cellulose by lytic polysaccharide monooxygenases (LPMOs) is crucial for efficient aerobic decomposition of plant biomass[3–6]. LPMOs are abundant in Nature and classified, based on their sequences, in the auxiliary activity (AA) families 9–11 and 13–17 of the Carbohydrate Active enZymes (CAZy) database[7]. LPMOs are mono-copper enzymes[4,5] that catalyze oxidative cleavage of glycosidic bonds in insoluble polysaccharides such as cellulose[5,6] and chitin[3], as well as in certain hemicelluloses[8,9]. LPMOs were first considered monooxygenases as the activity was shown to depend on the presence of molecular oxygen, but recent studies have demonstrated that $H_2O_2$ is the kinetically relevant co-substrate making these enzymes peroxygenases rather than monooxygenases[10–14]. The oxidative action of LPMOs disrupts the crystalline polysaccharide surface[15,16] thus promoting depolymerization by hydrolytic enzymes[3,17]. It is generally accepted that LPMOs are the C1 factor hypothesized by Elwyn Reese and co-workers in 1950[18] and that LPMOs explain why Eriksson et al. found, in 1974, that oxygen promotes biomass conversion by a fungal secretome[19].

[1]Faculty of Chemistry, Biotechnology and Food Science, Norwegian University of Life Sciences (NMBU), 1432 Ås, Norway. [2]Department of Biotechnology and Food Science, Norwegian University of Science and Technology (NTNU), 7491 Trondheim, Norway. [3]SINTEF Industry, Process Chemistry and Functional Materials, 0373 Oslo, Norway. [4]National Laboratory of Energy and Geology (LNEG), 1649-038 Lisboa, Portugal. ✉e-mail: vincent.eijsink@nmbu.no

LPMO catalysis was first thought to require delivery of two electrons, two protons and molecular oxygen per catalytic cycle in what would be a monooxygenase reaction (R-H + 2e⁻ + 2H⁺ + O₂ → R-OH + H₂O), whereas in the peroxygenase reaction, a reduced LPMO can catalyze multiple turnovers with $H_2O_2$ (R-H + $H_2O_2$ → R-OH + H₂O)[20]. A standard monooxygenase reaction set-up involves incubating the LPMO with substrate and a reductant under aerobic conditions and it has been shown that a wide variety of reducing compounds and reducing equivalent-delivering enzymes can drive LPMO reactions[4,21–27]. It is currently being debated whether observed monooxygenase reactions are in fact peroxygenase reactions that are limited by the in situ generation of $H_2O_2$ by LPMO-catalyzed or abiotic oxidation of the reductant (e.g., Bissaro et al.[28]). Importantly, like for other redox enzymes, high levels of $H_2O_2$ combined with low levels of substrate will lead to autocatalytic oxidative damage in the catalytic center of the enzyme[10,17,29]. $H_2O_2$-driven LPMO catalysis is a double-edged sword, enabling high enzymatic activity at the possible cost of enzyme inactivation.

Light represents an abundant and cheap source of energy that can be harvested by a photoredox catalyst to tailor $H_2O_2$ levels to enzymatic reactions[30,31]. Light-driven LPMO reactions were first described in 2016. Cannella et al.[32] showed that the activity of a fungal LPMO acting on amorphous cellulose (PASC) could be boosted dramatically by adding chlorophyllin, a photosynthetic pigment, and light, next to the reductant, ascorbic acid (AscA). Light-driven activity of a bacterial LPMO from *Streptomyces coelicolor* (*Sc*AA10C) on crystalline cellulose (Avicel) using irradiated vanadium-doped titanium dioxide (V-TiO₂) was demonstrated later the same year[33]. Both studies discussed molecular mechanisms for the observed LPMO activity, but neither considered light-induced formation of $H_2O_2$ from O₂ as the primary driver for LPMO activity, which, later, was shown to be the key driver of LPMO activity in these light-fueled reaction systems[23].

The impact of light on biomass conversion is of great interest, with repercussions spanning from the global carbon cycle to industrial biorefining. Light has been demonstrated to facilitate microbial decomposition of plant litter by increasing the accessibility of cell wall polysaccharides to enzymatic conversion[34–38]. Since secondary plant cell walls, the natural substrates of LPMOs, are rich in lignin, and since lignin is photoactive and can promote formation of $H_2O_2$[39,40], we hypothesized that light-driven redox processes involving lignin and LPMO activity can help explain the observed photofacilitation of biomass decomposition. Of note, possible effects of light may also be relevant for reactor design in industrial biorefining of lignocellulosic

biomass, since pretreated feedstocks that are subjected to enzymatic saccharification with LPMO-containing cellulolytic enzyme cocktails usually contain large amounts of lignin.

Here we report a detailed biochemical study of cellulose degradation by *Sc*AA10C, a well-studied model LPMO from the soil actinomycete *Streptomyces coelicolor*, using light-exposed lignin to fuel the LPMO reaction. We show that light-exposure of lignin has a large effect on LPMO activity and that this effect is driven by the ability of lignin to promote generation of $H_2O_2$. We also show that the necessary priming reduction of the LPMO may be achieved through direct interactions with polymeric lignin and that LPMOs, thus, can oxidize lignin. Using NMR spectroscopy, we demonstrate the impact of visible light on the lignin structure, revealing effects on olefinic structures. Next to providing insight into how lignin and light-exposed lignin affect LPMO activity, this study offers an alternative, enzyme-based explanation for the effect of light on biomass turnover in the biosphere.

## Results
### Photocatalytic hydrogen peroxide generation by lignin fuels LPMO activity on cellulose
Previous studies have demonstrated lignin's ability to fuel LPMO reactions and this was thought to reflect the ability of lignin to deliver the electrons needed by the LPMO to carry out a monooxygenase reaction[24,25,32,41]. To gain more insight into lignin's ability to fuel LPMO reactions and to assess the impact of light, we used a well-studied cellulose-active C1-oxidizing LPMO from *Streptomyces coelicolor* (*Sc*AA10C, also known as CelS2) and Avicel (i.e., crystalline cellulose) as substrate.

In the first set of experiments, we used commercially available kraft lignin to fuel solubilization of crystalline cellulose by *Sc*AA10C and we measured both LPMO product formation and the accumulation of $H_2O_2$ in reactions exposed to light (Fig. 1). As expected, oxidized cello-oligosaccharides were not generated in reactions lacking the LPMO (Fig. 1a). At the lower lignin concentration (0.9 g L⁻¹), the reaction without LPMO showed accumulation of $H_2O_2$, whereas the reaction with 75 nM or 500 nM LPMO showed almost identical linear progress curves for LPMO product formation and no accumulation of $H_2O_2$. This suggests that, under these conditions, the LPMO reaction was limited by generation of $H_2O_2$. At the higher lignin concentration (9 g L⁻¹), accumulation of $H_2O_2$ in the reaction without LPMO was much higher (Fig. 1b). In the reaction with only 75 nM LPMO, product formation stopped within the first hour (Fig. 1a) and $H_2O_2$ accumulated at a rate similar to the reaction without LPMO (Fig. 1b), indicating that the

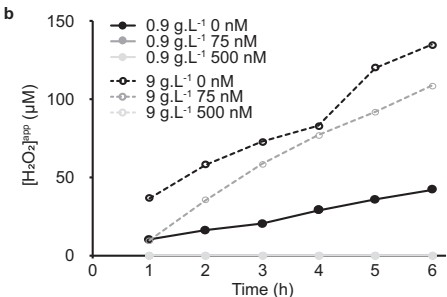

**Fig. 1 | LPMO-catalyzed depolymerization of cellulose using kraft lignin as photoredox catalyst.** The graphs show time-courses for the production of oxidized LPMO products (**a**) and apparent $H_2O_2$ levels (**b**) in photobiocatalytic reactions containing LPMO (*Sc*AA10C; 0, 75, or 500 nM; black, gray and light gray, respectively), substrate (Avicel, 10 g L⁻¹) and photoredox catalyst (kraft lignin; 0.9 or 9 g L⁻¹, closed symbols with solid lines and open symbols with dashed lines, respectively). All reactions were carried out in sodium phosphate buffer (50 mM, pH 7.0) at 40 °C under magnetic stirring and exposed to visible light ($I$ = 10% $I_{max}$, ~16.8 W cm⁻²). 50 μL aliquots were taken every hour and diluted with 50 μL water

prior to boiling for subsequent analysis of oxidized products (both soluble and insoluble) and quantification of $H_2O_2$. The data is reported as mean values from two individual experiments ($n$ = 2). The values showed 10% or less variation between replicates except for the reaction with 0.9 g L⁻¹ and 500 nM *Sc*AA10C where the deviations were less than 22% between replicates. No oxidized products were detected in reactions lacking LPMO (**a**) and $H_2O_2$ only accumulated in reactions without LPMO regardless of the lignin concentration except for the reaction with 9 g L⁻¹ lignin and 75 nM LPMO (**b**) (see text for an explanation). Reactions in the dark showed much lower product levels, as shown in Fig. 2.

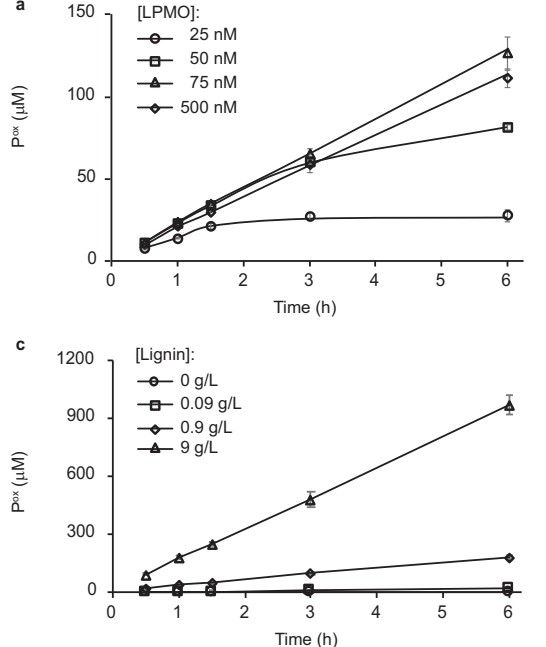

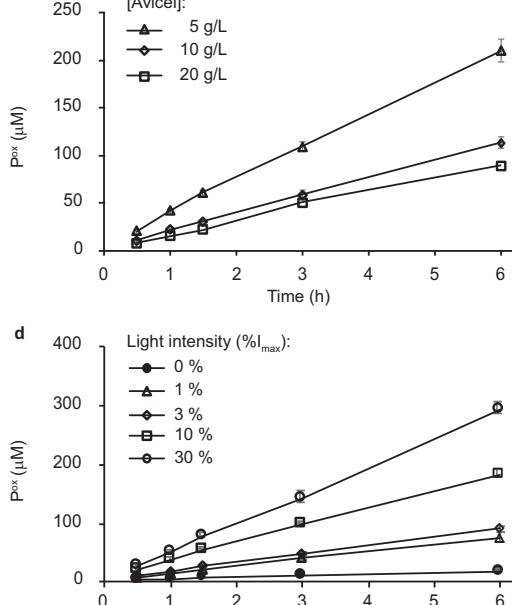

**Fig. 2 | Influence of the LPMO, Avicel, and lignin concentrations and light intensity on LPMO-catalyzed solubilization of cellulose.** The graphs show time-courses for the release of aldonic acid products in reactions with varying **a** LPMO concentration, **b** Avicel concentration, **c** kraft lignin concentration, and **d** light intensity. The values of the varied reaction parameter and the symbols used to discriminate different conditions are explained in the graphs. All reactions were carried out in sodium phosphate buffer (50 mM, pH 7.0) at 40 °C under magnetic stirring with exposure to visible light (10% $I_{max}$, -16.8 W cm$^{-2}$ unless otherwise specified), and contained LPMO (ScAA10C, 0.5 μM), Avicel (10 g L$^{-1}$), and lignin (0.9 g L$^{-1}$), unless otherwise specified. Before quantification of soluble oxidized products, solubilized cello-oligosaccharides were hydrolyzed by TfCel6A to convert LPMO products with varying degree of polymerization (DP) to a mixture of DP 2 and 3 [GlcGlc1A, (Glc)2Glc1A], the amounts of which were summed up to yield the concentration of oxidized sites. The concentration of oxidized sites is reported as the mean value from the three independent experiments and error bars show ±s.d. (n = 3).

LPMO had been inactivated due to an overload of H$_2$O$_2$[23,42,43]. To demonstrate enzyme inactivation, three separate reactions identical to the 9 g L$^{-1}$ lignin, 75 nM LPMO reaction of Fig. 1 were set up and after one hour, substrate, enzyme and substrate, or a reductant and substrate were added. Only the reaction to which fresh enzyme was added showed resumed LPMO activity (Supplementary Fig. 1), confirming that, indeed, enzyme inactivation had occurred. On the other hand, 500 nM LPMO was sufficient to productively convert all H$_2$O$_2$ generated during the course of the 6 h reaction with 9 g L$^{-1}$ lignin into oxidized cello-oligosaccharides and no H$_2$O$_2$ accumulation was observed in this reaction (Fig. 1). Consequently, product formation in the reaction with 9 g L$^{-1}$ lignin and 500 nM LPMO was much faster than in any of the other reactions.

While Fig. 1 shows that there is a clear correlation between the amount of H$_2$O$_2$ generated in the reaction system and LPMO activity, there is a marked difference between the H$_2$O$_2$ levels generated in absence of LPMO (Fig. 1b) and the amount of oxidized product formed in LPMO-containing reactions (Fig. 1a). If the apparent H$_2$O$_2$ levels in Fig. 1b equal the true levels and if one accepts the premise that access to H$_2$O$_2$ limits the LPMO reaction, H$_2$O$_2$ levels in the reaction without LPMO and LPMO product levels should be similar. One potential explanation resides in the HRP/Amplex Red assay used to determine H$_2$O$_2$ levels. Kraft lignin serves as substrate for HRP, which will suppress the Amplex Red signal. This effect was, however, compensated for since all H$_2$O$_2$ standard curves used to determine H$_2$O$_2$ accumulation with the HRP/Amplex Red assay contained the same lignin concentration as the reaction being analyzed. Another explanation lies in the abiotic consumption of H$_2$O$_2$ due to abiotic reactions with lignin[44]. The levels of H$_2$O$_2$ measured in the absence of the LPMO are the net result of formation (i.e., oxidation of lignin by O$_2$) and degradation (i.e., oxidation of lignin by H$_2$O$_2$), both of which may be dependent on light, as has been shown for a different photoredox catalyst[45]. Since LPMOs

in presence of substrate have high affinity for H$_2$O$_2$ ($K_m$ values in the low micromolar range)[11,29,43] it is conceivable that the LPMO peroxygenase reaction outcompetes consumption of H$_2$O$_2$ through reactions with lignin, which would explain the discrepancy between apparent H$_2$O$_2$ measured and LPMO product levels. A control experiment indicated that, indeed, H$_2$O$_2$ consumption by the LPMO is faster than abiotic H$_2$O$_2$ consumption (Supplementary Fig. 2).

To further understand the lignin/light/LPMO system, each reaction component in a standard reaction with ScAA10C (0.5 μM), Avicel (10 g L$^{-1}$), lignin (0.9 g L$^{-1}$), and light (I = 10% $I_{max}$, corresponding to -16.8 W cm$^{-2}$) was varied. In these, and subsequent, experiments only soluble LPMO products were quantified. Further reduction of the LPMO concentration to below 75 nM showed that the LPMO became limiting at lower concentrations (Fig. 2a). At 50 nM LPMO, product formation appeared to level off between 3 and 6 h, and further reducing the LPMO concentration to 25 nM resulted in cessation of product formation after 90 min due to enzyme inactivation (Fig. 2a).

Increasing the Avicel concentration led to a decrease in LPMO activity (Fig. 2b). While this may seem counterintuitive, it has been shown that higher Avicel concentrations attenuate more photons[42] which would reduce lignin-catalyzed H$_2$O$_2$ formation. Control reactions without enzyme showed that, indeed, the production of H$_2$O$_2$ in light-exposed reactions with a fixed amount of lignin is inversely correlated with the Avicel concentration (Supplementary Fig. 3). As for the lignin concentration, a clear dose-response effect was already visible in the data of Figs. 1 and 2c shows that further lowering of the lignin concentration leads to less LPMO activity, confirming the dose-response relationship. Figure 2d shows a clear dose-response effect for the light and shows that the reaction with the standard amount of light used here (I = 10% $I_{max}$) is one order of magnitude faster than a reaction in the dark. No LPMO activity was detected in absence of lignin (Fig. 2c). Taken together, the results displayed in Figs. 1 and 2

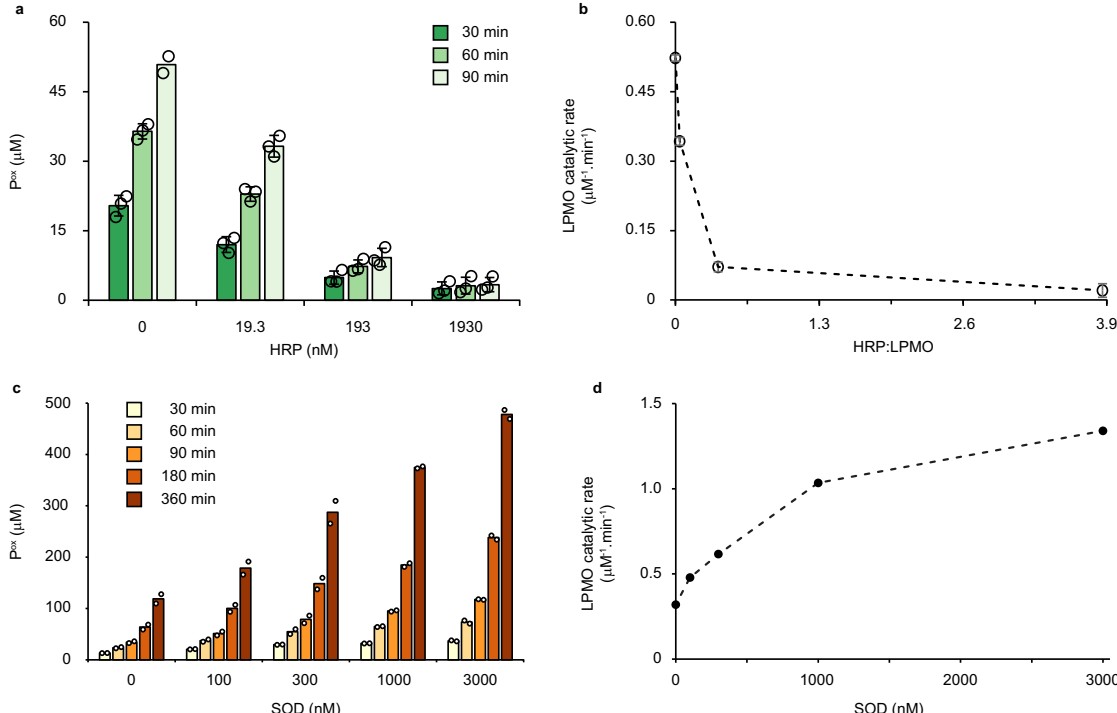

**Fig. 3 | Probing the role of reactive oxygen species in the light/lignin/LPMO system.** The graphs show time-courses for the formation of soluble oxidized products (**a**, **c**) and the corresponding apparent catalytic rates (**b**, **d**) for reactions with Avicel (10 g L$^{-1}$), *Sc*AA10C (0.5 μM), kraft lignin (0.9 g L$^{-1}$), and light-exposure in the presence of varying amounts of horseradish peroxidase (HRP) (**a**, **b**) and superoxide dismutase (SOD) (**c**, **d**). The varying colors in panels **a** and **c** indicate different time points of the reaction, as explained in the graphs. The rates shown in **b** were derived from linear regression analysis using all three time points in **a** with $R^2 > 0.99$ for all reactions with 0, 19.3, 193 nM HRP except for one replicate with 193 nM with $R^2 > 0.93$. For the reactions with 1930 nM HRP the product levels were very low and showed larger variability as these levels were close to the detection

limit of the analytical method. The rates in **d** were derived using linear regression analysis for all time points displayed in **c** and all reactions gave progress curves with $R^2 > 0.99$. All reactions were carried out in sodium phosphate buffer (50 mM, pH 7.0) at 40 °C, under magnetic stirring and exposed to visible light ($I = 10\%$ $I_{max}$, -16.8 W cm$^{-2}$). Before quantification of soluble oxidized products, solubilized cello-oligosaccharides were hydrolyzed by *Tf*Cel6A to convert LPMO products with varying degree of polymerization (DP) to a mixture of DP 2 and 3 [GlcGlc1A, (Glc) 2Glc1A], the amounts of which were summed up to yield the concentration of oxidized sites. The data presented are mean values derived from three (**a**, **b**) or two (**c**, **d**) independent experiments; error bars show ±s.d. (**a**, **b**; $n = 3$).

demonstrate that combining lignin and light enables fine-tuning of LPMO reactions and that increased LPMO activity correlates with conditions that favor H$_2$O$_2$ production. Preliminary experiments with fungal cellulose-active AA9 LPMOs showed that also in this case lignin-driven LPMO activity was boosted by visible light (Supplementary Fig. 4).

To demonstrate that light-driven H$_2$O$_2$ generation fuels the LPMO reaction, competition experiments were performed with increasing amounts of horseradish peroxidase (HRP). No additional substrate for HRP was needed as the soluble lignin used in these reactions is a suitable substrate for this enzyme. The reaction catalyzed by 0.5 μM LPMO was increasingly inhibited by increasing amounts of HRP (Fig. 3a). Plotting the rate of LPMO catalytic activity against the HRP concentration showed more than 85% inhibition of LPMO activity with 193 nM HRP and almost complete inhibition, >97% inhibition, with 1930 nM HRP (Fig. 3b). These experiments clearly show that the LPMO reaction is fueled by the H$_2$O$_2$ generated from light-irradiated lignin.

Two recent studies have demonstrated H$_2$O$_2$ generation by light-exposed lignin, which may be the result of two single-electron reductions of O$_2$ leading to O$_2^{\cdot-}$ and then H$_2$O$_2$, or of a one-step, two-electron reduction of O$_2$ to H$_2$O$_2$[39,40]. Of note, the superoxide radical can likely act as reductant for the LPMO[23,46]. To assess possible formation of superoxide we carried out reactions with superoxide dismutase (SOD), which converts superoxide to H$_2$O$_2$ and O$_2$. Adding increasing amounts of SOD (0–3000 nM) to an irradiated reaction with lignin (0.9 g L$^{-1}$), Avicel (10 g L$^{-1}$), and *Sc*AA10C (0.5 μM) led to a near four-fold increase in the LPMO rate (Fig. 3c, d), showing that superoxide was

indeed generated from light-exposed lignin and that access to H$_2$O$_2$ limits LPMO activity in these conditions.

## LPMO reduction by lignin

Superoxide and lignin have both been suggested as competent reducing agents for LPMOs[23–25]. To create insight into the role of lignin in LPMO reduction, we assessed the ability of lignin to reduce the LPMO using stopped-flow kinetic measurements. We first attempted to do so with *Sc*AA10C, but for this LPMO the combination of a weak signal and signal quenching by lignin prevented the determination of rates from the kinetic traces (see Supplementary Fig. 5 for data and further discussion). Changing from the cellulose-active *Sc*AA10C to the chitin-active *Sm*AA10A, with a stronger fluorescence signal, allowed proper determination of lignin oxidation rates (Supplementary Fig. 5). Of note, a control experiment showed that, just as cellulose degradation by *Sc*AA10C, chitin degradation by *Sm*AA10A was boosted by light-exposed lignin (Supplementary Fig. 6).

To rule out that LPMO reduction was caused by small phenolic or other low molecular weight compounds present in the commercial kraft lignin preparation, we measured LPMO reduction both with native kraft lignin and dialyzed kraft lignin. Such a dialysis step is often performed when studying lignin peroxidases to remove traces of Mn$^{2+}$[47]. The effect of lignin dialysis was minimal, both for light-driven (aerobic) cellulose oxidation by *Sc*AA10C and, importantly, for (anaerobic) reduction of *Sm*AA10A (Fig. 4 and Supplementary Fig. 5). Figure 4a shows that reactions with native and dialyzed kraft-lignin generated similar levels of oxidized products during a 6 h reaction with

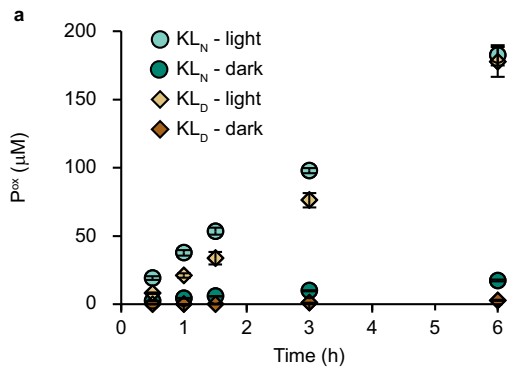

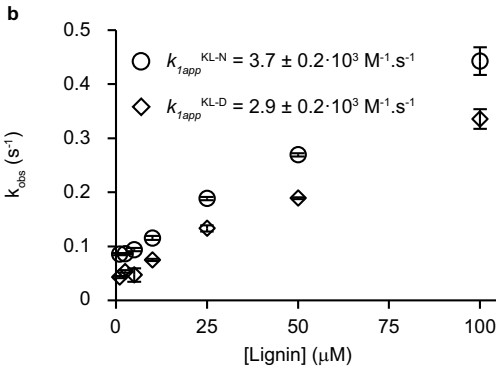

**Fig. 4 | Lignin-driven ScAA10C-catalyzed solubilization of cellulose and reduction kinetics for SmAA10A. a** The figure shows time courses for the formation of solubilized oxidized products by ScAA10C (0.5 μM) in reactions with native (KL$_N$; circles) or dialyzed (KL$_D$; diamonds) kraft lignin (0.9 g L$^{-1}$) and Avicel (10 g L$^{-1}$) in sodium phosphate buffer (50 mM, pH 7.0) at 40 °C with magnetic stirring, in the dark (darker color) or when irradiated by white light (lighter color; $I = 10\%$ $I_{max}$, -16.8 W cm$^{-2}$). Before quantification of soluble oxidized products, solubilized cello-oligosaccharides were hydrolyzed by TfCel6A to convert LPMO products with varying degree of polymerization (DP) to a mixture of DP 2 and 3 [GlcGlc1A, (Glc)2Glc1A], the amounts of which were summed up to yield the concentration of oxidized sites. **b** The figure shows the observed pseudo-first-order constants, k$_{obs}$, for reduction of SmAA10A-Cu(II) as a function the kraft lignin concentration, derived from the fluorescence traces shown in Supplementary

Fig. 3a, b. Kraft lignin concentrations were calculated based on an average molecular mass (provided by the supplier) of 10,000 g/mol for both lignin preparations; since the average mass of the dialyzed lignin is expected to be somewhat higher, compared to the native lignin, the second order rate constant for the dialyzed lignin is underestimated. SmAA10A-Cu(II) (10 μM) was anaerobically mixed with varying concentrations of native (KL$_N$; circles) and dialyzed (KL$_D$; diamonds) kraft lignin, and the change in fluorescence was monitored as a function of time. The reactions were carried out in sodium phosphate buffer (50 mM, pH 7.0) at 25 °C. Data were fit to single exponential functions to give observed rate constants (k$_{obs}$) at each lignin concentration. The apparent second order rate constant $k_{1app}$$^{lignin}$ was determined from linear regression using the reported data points and displayed an $R^2 > 0.99$. The data in **a** and **b** are reported as mean values from three independent experiments and the error bars show ± s.d. ($n = 3$).

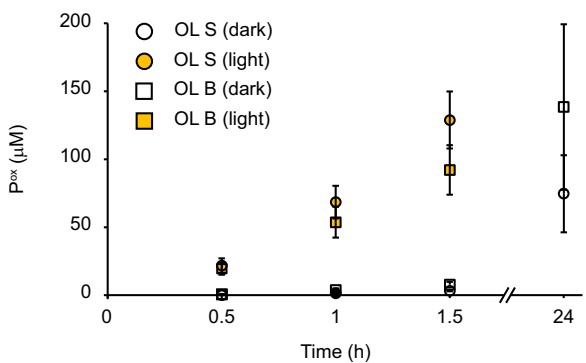

**Fig. 5 | LPMO-catalyzed depolymerization of cellulose using organosolv lignin as photoredox catalyst.** The graph shows time courses for the production of oxidized products in photobiocatalytic reactions containing ScAA10C (500 nM), Avicel (10 g L$^{-1}$), and organosolv lignin (OL) from spruce (S; circles) or birch (B; squares) (2.5 g L$^{-1}$). All reactions were carried out in sodium phosphate buffer (50 mM, pH 6.0) at 40 °C under magnetic stirring and exposed (orange symbols) or not (white symbols) to visible light ($I = 10\%$ $I_{max}$, -16.8 W cm$^{-2}$). The light-exposed reactions were incubated for 1.5 h while the dark reactions were incubated for 24 h. Before quantification of soluble oxidized products, solubilized cello-oligosaccharides were hydrolyzed by TfCel6A to convert LPMO products with varying degree of polymerization (DP) to a mixture of DP 2 and 3 [GlcGlc1A, (Glc)2Glc1A], the amounts of which were summed up to yield the concentration of oxidized sites. The data is presented as mean values obtained from three independent experiments and error bars show ±s.d. ($n = 3$). OL was prepared as a stock suspension (25 g L$^{-1}$) in water, and thoroughly mixed prior to adding lignin to the reaction vials.

light-exposure. For reactions in the dark, the dialyzed lignin resulted in lower LPMO activity compared to the already slow reaction with native kraft lignin (Fig. 4a). It is conceivable that under these conditions, the presence of rapidly diffusing low molecular weight reductants has a notable impact on the (low) rate of in situ H$_2$O$_2$ generation that drives the reaction. Figure 4b shows that anaerobic reduction of SmAA10A, and, thus oxidation of lignin, happens with similar second order rate

constants, $k_{1app}$$^{lignin}$, of $3.7 \times 10^3$ M s$^{-1}$ and $2.9 \times 10^3$ M s$^{-1}$, for non-dialyzed and dialyzed kraft lignin, respectively. These results demonstrate that the copper site of LPMOs can directly interact with and oxidize a high molecular weight lignin polymer. Although no reliable rates could be obtained for the cellulose-active ScAA10C, the data suggested that reduction of this enzyme was slower than reduction of SmAA10A (Supplementary Fig. 5).

## Studies with other lignin types
Kraft lignin is produced from kraft pulping of wood to separate cellulose from hemicellulose and lignin using sodium hydroxide and sodium sulfide. This process generates a modified and condensed lignin structure with an increase in phenolic groups and recalcitrant C-C and C-O bonds, and a reduced number of less recalcitrant β-O-4 bonds, compared to native lignin[48,49]. To assess the impact of lignin type on light-enhanced LPMO activity, we performed experiments similar to those reported above in which the soluble kraft lignin was replaced by insoluble organosolv lignin obtained from either spruce or birch. Figure 5 shows that light-exposure drastically enhanced the ability of insoluble organosolv lignin to fuel the LPMO reaction, similar to what was observed with kraft lignin.

## Light-induced structural changes of lignin
The boosting effect of light on lignin-driven LPMO-catalyzed oxidation of cellulose originates from the ability of lignin to photocatalytically reduce O$_2$ to O$_2$$^{\cdot-}$ and H$_2$O$_2$. Ring-conjugated double bonds, like those found in the cinnamyl alcohol building blocks, in β−1 stilbenes, and carbonyl moieties are known lignin structures that absorb light[50]. Irradiating Cα-carbonyls in lignin with UV-light leads to excited state carbonyls which may abstract phenolic hydrogens to yield phenoxyl radicals, but visible light does not provide the energy needed to excite Cα-carbonyls[51]. Recently, it has been proposed that the Cα-OH moieties of β-O-4 bonds in lignin are involved in O$_2$ reduction to H$_2$O$_2$, resulting in the conversion of Cα-OH to Cα = O[40]. Supporting this notion, Kim et al. showed photocatalytic reduction of O$_2$ to H$_2$O$_2$ using a model lignin dimer, guaiacylglycerol-β-guaiacyl ether, which contains two guaiacyl units linked together via a β-O-4 bond and harbors a

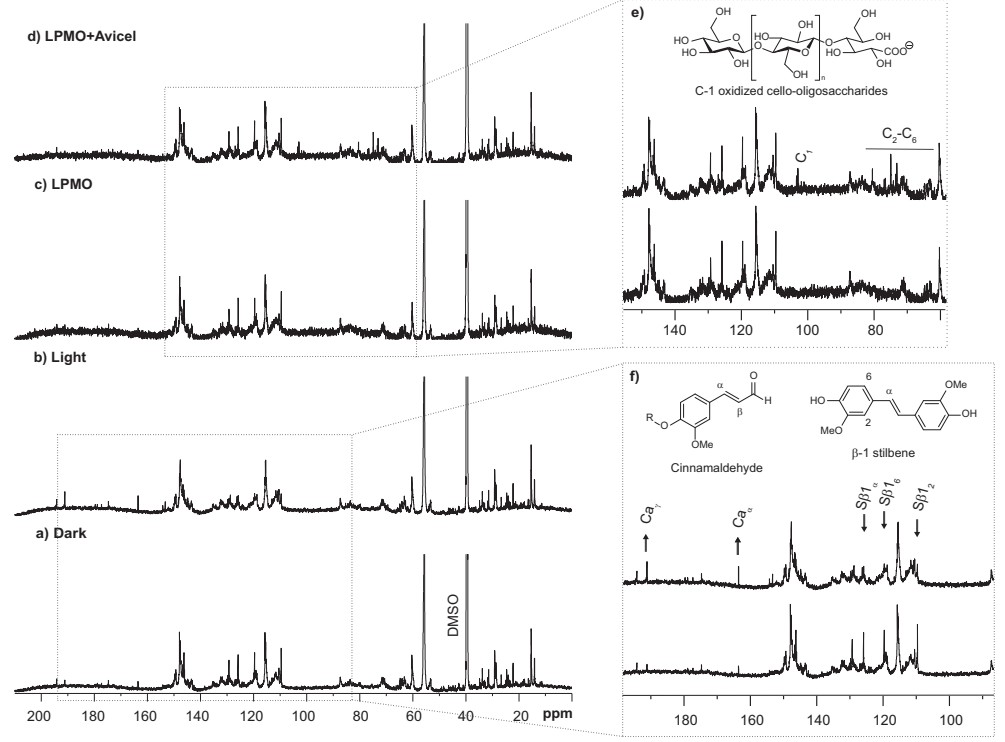

**Fig. 6 | Light-induced and LPMO-induced changes in organosolv spruce lignin assessed by 1D carbon NMR spectroscopy.** The panels show the spectra obtained for organosolv lignin from spruce (10 g L$^{-1}$) incubated for 24 h in the dark (**a**), with light-exposure ($I$ = 10% $I_{max}$, corresponding to -16.8 W cm$^{-2}$) (**b**), in the dark with *Sc*AA10C (500 nM) (**c**), or in the dark with *Sc*AA10C (500 nM) and Avicel (10 g L$^{-1}$) (**d**). Regions of the spectra displaying differences related to treatment with light (**f**) or an LPMO (**e**) are shown in the panels to the right. There were no detectable differences in the parts of the spectra that are not shown in panels **e** and **f**. All reactions were performed in sodium phosphate buffer (50 mM, pH 6.0) at 40 °C with magnetic stirring. The NMR samples were prepared by dissolving either ~40 mg for light-treated lignin (**a**, **b**, **f**) or ~20 mg for LPMO-treated lignin (**c**, **d**, **e**) in

480 μL DMSO-d$_6$ (99.96 atom % D) and the carbon spectrum was recorded at 25 °C on an 800 MHz instrument. To account for the differences in lignin concentration the intensity of all spectra was adjusted to be equal for the signal at -28 ppm. Identified chemical moieties are based on partial assignment using $^1$H–$^{13}$C-HSQC and previous values reported in the literature. Signals from β-1 stilbene (Sβ1$_\alpha$, Sβ1$_2$, and Sβ1$_6$)[49], cinnamaldehyde (Ca$_\alpha$ and Ca$_\gamma$)[49, 53], and C-1 oxidized cello-oligosaccharides [C$_1$, C$_2$-C$_6$, where the number refers to the ring carbon for the monosaccharide[54]] are indicated. Changes in the abundance of selected chemical moieties are indicated with an up arrow for increase and a down arrow for decrease upon light treatment, and R indicates further coupling to the lignin polymer (**f**).

Cα-OH. The Cα-OH was shown to be photocatalytically oxidized to Cα = O with concomitant H$_2$O$_2$ formation, whereas the lignin monomers coniferyl alcohol and sinapyl alcohol were shown unable to photocatalytically reduce O$_2$ to H$_2$O$_2$[40]. When we employed the same lignin dimer in light-exposed LPMO reactions we did not observe H$_2$O$_2$ formation nor LPMO activity. Thus, we searched for other modifications (oxidations) in the lignin that are promoted by light exposure.

NMR spectroscopy was used to qualitatively investigate light-induced and LPMO-induced changes in the lignin structures directly. All lignins were incubated for 24 h with or without exposure to visible light ($I$ = 10% $I_{max}$, corresponding to -16.8 W cm$^{-2}$). For kraft lignin, light-exposure resulted in a decrease in the signal corresponding to hydroxyl groups (Supplementary Fig. 7), which could be due to generation of phenoxyl radicals (i.e., oxidation of phenolic hydroxyl groups) that radically couple with other parts of the lignin structure. More extensive analyses were done with the organosolv lignins. For organosolv lignin from both birch and spruce, the light treated sample showed an increase in cinnamaldehyde end groups (see Fig. 6a, b, f for spruce and Supplementary Fig. 8a, b, f for birch; more details in Supplementary Figs. 9 and 10), a decrease in carbon-carbon double bonds (Supplementary Figs. 9a and 10a), and, in the case of spruce, a notable decrease in β−1 stilbene signals (SB1$_\alpha$, SB1$_2$, SB1$_6$ in Fig. 6f). Overall, the spectra of light-exposed organosolv lignin showed a decrease in signals associated with olefins, accompanied by an increase in aldehyde signals (Supplementary Figs. 9 and 10). The decrease in olefinic signals

and the concomitant increase in aldehydes are consistent with light-induced oxidation of ring-conjugated olefins[50].

Given that *Sc*AA10C oxidizes lignin and that organosolv lignin sustains slow cellulose solubilization by *Sc*AA10C in the dark, we attempted to measure changes in the organosolv lignin structure following reactions in the dark with LPMO, in the absence or presence of Avicel. Based on 1D carbon NMR, the lignin structure seemed unaffected by the LPMO regardless of the presence of Avicel (Fig. 6c–e, Supplementary Fig. 8c–e). When Avicel was included, the presence of soluble C-1 oxidized cello-oligosaccharides (Fig. 6e, Supplementary Figs. 8e, 9c, and 10c) was clearly detectable, showing that the LPMO was active. It should be noted that the spectra for LPMO-treated lignin have a higher signal-to-noise ratio compared to the spectra for light-treated lignin due to a 2-fold lower lignin concentration leading to ~4-fold lower sensitivity.

1D proton NMR of the treated organosolv lignins showed that protons of the hydroxyl groups in light-treated lignin occur at a higher chemical shift meaning that they are on average more deshielded compared to dark-incubated lignin. In contrast, addition of the LPMO resulted in hydroxyl protons becoming more shielded, as shown by a lower chemical shift (Supplementary Fig. 11). The degree of shielding may be interpreted as the degree of hydrogen bonding, as hydroxyl groups are strongly deshielded by hydrogen bonds[52]. These changes were observed for both the spruce and the birch lignin and suggest that light-driven oxidation and LPMO-catalyzed oxidation of lignin

have different chemical consequences. Oxidation of ring-conjugated olefins, promoted by light, could lead to some depolymerization of the lignin (as also suggested by the increase in cinnamaldehyde end groups; Fig. 6), resulting in increased hydrogen bonding and deshielded hydroxyl groups. On the other hand, LPMOs will oxidize hydroxyl groups[22], which could lead to radical formation and increased polymerization. It is not surprising that, apart from the observed changes in hydrogen bonding of the hydroxyl protons, no effects of LPMO treatment on the lignin structure could be detected, given that a reduced LPMO can catalyze multiple peroxygenase reactions and that, thus, oxidation of lignin by the LPMO may be much less frequent than the light-promoted oxidations that generate $H_2O_2$.

### Probing for a possible role of water oxidation

It has been claimed, recently, that lignin may photocatalytically oxidize $H_2O$ to $H_2O_2$ and $O_2$[40], which would mean that the formation of $H_2O_2$ by irradiated lignin does not depend on $O_2$, and that irradiated lignin should be able to fuel the LPMO reaction under anaerobic conditions. To assess this possibility, anaerobic experiments with ScAA10C and Avicel were performed, in the presence of lignins (soluble kraft lignin and insoluble organosolv lignin from spruce) or ascorbic acid. The reaction containing only AscA should not lead to any product formation in true anaerobic conditions whilst a control reaction containing AscA and $H_2O_2$ should generate oxidized products.

Chromatographic analysis of reaction mixtures after 22 h of incubation under anaerobic conditions, showed that all three reactions without added $H_2O_2$ had generated identical, low amounts of oxidized products, whereas, as expected, product levels were higher in the reaction with added $H_2O_2$ (Supplementary Fig. 13). The similar and low product levels in the reactions without added $H_2O_2$, regardless of the reductant (AscA or lignin), indicate that all reactions were limited by the same factor, which must be traces of $O_2$. The chromatographic analysis shows that, if water oxidation was happening at all in the reaction set-ups used here, this process must have been very slow, since neither kraft lignin nor organosolv spruce lignin were able to promote anaerobic LPMO activity above the level reached in the anaerobic reaction with AscA. We did these experiments in $H_2^{18}O$ and used $H_2^{18}O_2$ in the control reaction with hydrogen peroxide, because such an approach in principle could provide additional evidence for (the absence of) water oxidation, as explained in the legend of Supplementary Fig. 13. Unfortunately, due to the presence of lignin, the quality of MALDI-TOF MS spectra was too low to provide additional support for the conclusions drawn from chromatographic product analysis.

## Discussion

Biotic degradation of recalcitrant carbohydrates in plant litter is promoted by sunlight. This effect is believed to stem from photodegradation of lignin in secondary plant cell walls, which would increase the availability of cell wall carbohydrates for enzymatic degradation[34–36,38]. LPMOs are key to aerobic solubilization of cellulose and other polysaccharides[55,56] from plant cell walls and, in the present study, we show that the impact of light on biomass degradation may relate to the activity of these enzymes. We show that irradiation of lignin promotes lignin oxidation and formation of $H_2O_2$, which fuels the LPMO reaction. Notably, abiotic generation of $H_2O_2$ in the biomass may also promote the activity of other biomass-converting and $H_2O_2$-consuming enzymes, for example lignin peroxidases.

This study provides further evidence for $H_2O_2$-driven LPMO activity and adds to the notion that LPMOs are peroxygenases, and that the monooxygenase activity of these enzymes, if existing at all, is of minor importance, kinetically. We demonstrate that LPMO activity is improved in conditions generating higher $H_2O_2$ levels and is inhibited by HRP, supporting the notion that the LPMO reaction is $H_2O_2$-dependent. Since LPMOs are susceptible to autocatalytic

inactivation[10,57], as also demonstrated here, in Fig. 1 and Supplementary Fig. 1, regulating the amount of $H_2O_2$ available to the LPMO is important. The use of lignin and light not only offers a cheap and abundant source of reducing power for LPMO reactions, but could also be used to obtain better control and regulation, as previously shown for light-driven LPMO reactions with chlorophyllin[32,42,58]. It should be noted that the use of light to control LPMO activity in commercial bioreactors operating at high dry matter concentrations with for instance lignocellulose will be challenging as light is attenuated in reaction slurries. Still, light will penetrate to some extent and it is thus worth noting that the present results suggest that the outcome of lignocellulose saccharification experiments with LPMO-containing cellulase cocktails may depend on the vessel type (glass or steel) and the light conditions in the laboratory or the industrial plant. These light attenuation issues will not apply in light/lignin fueled reaction with other $H_2O_2$-dependent enzymes, for example the oxyfunctionalization of hydrocarbons recently reported by Kim et al.[40].

LPMO catalysis depends on reducing equivalents that are needed to bring the enzyme in its reduced, catalytically competent state. Since a once reduced LPMO can catalyze multiple peroxygenase reactions[14,17,59] and since most LPMO reactions likely are limited by available $H_2O_2$, the amount of LPMO reduction needed to maintain optimal reaction speed is somewhat unclear but is certainly much lower than the need for in situ generation of $H_2O_2$. We show here that LPMOs can oxidize polymeric lignin directly to recruit electrons and do so at an appreciable rate. The rates determined in our stopped-flow experiments are one order of magnitude lower than those observed for lignin oxidation by manganese peroxidase[60], between two and three orders of magnitude lower than the most efficient lignin peroxidases[61], and two orders of magnitude lower than LPMO reduction by one of the most efficient small molecule reductants, AscA[12].

While photoyellowing and photobleaching of lignin are well-known phenomena[50], and studies on the impact of visible light on lignin model compounds and lignin combined with (non-lignin) photoredox catalysts have been reported[62,63], to our knowledge not much is known about the structural modifications that may occur when polymeric lignin is exposed to visible light ($\lambda = 400–700$ nm). Our NMR analysis reveals that visible light-exposure of lignin results in oxidation of ring-conjugated carbon-carbon double bonds with a concomitant increase in cinnamaldehyde end groups (Fig. 6, Supplementary Figs. 8–11). Following light-exposure, the lignin hydroxyl groups experience an increase in hydrogen bonding, an effect that is opposite of what was found when the lignin was incubated in the presence of an LPMO, in the dark. This indicates that light-induced oxidation of lignin and LPMO-catalyzed lignin oxidation are distinct reactions

Importantly, while the structural studies of lignin show effects of both irradiation and LPMO action and clearly point at the chemical processes involved, further studies are needed to fully unravel structural changes in lignin. We used the highest practical sample concentrations in the NMR analyses, to maximize sensitivity. The complexity and heterogeneity of the lignin structures requires high sensitivity, while achieving complete dissolution of samples is challenging. It is likely that the structural changes in lignin observed in this study only provide part of the picture, due to low signal-to-noise ratios, particularly for the 1D carbon spectra. Of note, the apparent lack of an effect of LPMO treatment on the 1D carbon spectra of lignin (Fig. 6 and Supplementary Fig. 8) could to some extent be due to the lower signal-to-noise ratio in these spectra (compared to the spectra obtained in the experiments with light). Thus, we cannot fully exclude that LPMO action also leads to lignin oxidations similar to those occurring upon treatment with light. Further in-depth studies of treated and untreated lignin are needed to unravel the full impact of light and LPMO action of lignin. Such studies may eventually allow the determination of quantitative correlations between the degree of lignin oxidation, the amount of hydrogen peroxide produced and LPMO activity. Of note,

revealing such correlations would require accurate quantitative detection of all LPMO products and hydrogen peroxide levels under relevant conditions, which is challenging for reactions with lignin.

The present findings show that LPMO reactions can be fueled by light-exposed lignin and may have wide implications for how we understand biological processes related to biomass conversion in Nature. Lignin is abundant in plant biomass, which could make many processes involving biomass light sensitive. Interestingly, LPMO action was recently shown to be a major contributor to the infectivity of the potato pathogen *Phytophtora infestans*[64] and one may wonder if infectivity is affected by light. On another note, our findings suggest that changes in access to light may contribute to the well-known impact of tillage regimes on the turnover and sequestration of organic matter in soil[65]. It would be of interest to investigate whether the interplay between light, redox-active structural components, and enzymes such as LPMOs has had an impact on the (co-)evolution of lignin-rich materials and the enzyme systems that degrade these. While these are interesting possible implications and while the impact of light on biomass conversion in Nature is indisputable, the magnitude and relative importance of light/lignin-fueled catalysis by LPMOs and other $H_2O_2$-dependent biomass degrading enzymes remains to be established. No matter the width and magnitude of these implications, the present study provides important insight into the complex roles of lignin and light in Nature and the catalytic potential of LPMOs.

## Methods

### Materials

The crystalline cellulose used in this study was Avicel PH-101 (50 μm particles; Sigma-Aldrich). A 10 mM stock solution of AmplexRed (Thermo Fisher Scientific) was prepared in DMSO, aliquoted, and stored at −20 °C in the dark. Aliquots were thawed in the dark for 10 min before use and were used only once. Lignin stock solutions were prepared fresh in water each day in aluminum foil wrapped tubes and kept on ice. Kraft lignin, with an average molecular mass of 10 000 g/mol, was purchased from Sigma-Aldrich (Product number: 471003) and stored at room temperature in the dark. Dialyzed kraft lignin was prepared by dialyzing ~25 mL of a saturated kraft lignin solution against 5 L of ultrapure Milli-Q treated water overnight three times, in the dark, using a Spectra/Por® membrane with a MWCO of 3500 Da, after which the material was freeze-dried (Supplementary Fig. 12).

Organosolv lignins were obtained from spruce and birch. Debarked knife-milled wood (<2 mm) was used as feedstocks for organosolv treatments conducted in a 600 mL stirred high-pressure reactor (Parr) using 50 wt % aqueous ethanol as solvent and a biomass content in the reactor of 10 wt %. The wood suspensions were kept at 190 °C for 90 min or 120 min, for birch or spruce, respectively. After the treatment, the slurries were separated using a hydraulic press (Sotel) and the liquid phase was vacuum filtered (Whatman filter paper no.1). Lignin precipitation was performed by diluting the organosolv hydrolysates with water (1:4, w/w). Precipitation experiments were conducted at room temperature, with magnetic stirring for 2 h. After that, the suspension was centrifuged for 30 min at 12,000 g. Supernatants were discarded and lignin was dried at 45 °C for at least 48 h. Stock suspensions of organosolv lignins for photobiocatalytic LPMO reactions were suspended in water, not in DMSO or alcohols as these solvents may act as radical scavengers and/or sacrificial electron donors.

### Enzymes

The model enzyme, *Sc*AA10C (UniProt ID Q9RJY2 [https://www.uniprot.org/uniprotkb/Q9RJY2/entry]) from *Streptomyces coelicolor*, was recombinantly produced and purified as previously described using anion exchange chromatography (HiTrap DEAE FF, GE Healthcare) followed by size exclusion chromatography (HiLoad 16/60

Superdex 75, GE Healthcare)[66], copper-saturated with three-fold molar excess Cu(II)SO$_4$[67], and desalted using a PD MidiTrap column [G-25, GE Healthcare][68] with buffer exchange to sodium phosphate (25 mM, pH 6.0). *Sm*AA10A (UniProt ID O83009) was produced and purified as previously described using chitin affinity chromatography (Chitin resin, New England Biolabs)[69], copper-saturated similarly to *Sc*AA10C, and stored in the same buffer. *Ta*AA9A (UniProt ID G3XAP7) was recombinantly produced and purified as described elsewhere using hydrophobic interaction chromatography (HiTrap Phenyl FF, GE Healthcare)[70] and copper-saturated prior to size-exclusion chromatography (HiLoad 16/60 Superdex 75, GE Healthcare)[71]. *Nc*AA9F (NCU03328; UniProt ID Q1K4Q1) was recombinantly produced and purified as described elsewhere[72] using hydrophobic interaction chromatography (HiTrap Phenyl FF, GE Healthcare) and anion exchange chromatography (HiTrap DEAE FF, GE Healthcare), and copper-saturated prior to size-exclusion chromatography (HiLoad 16/60 Superdex 75, GE Healthcare). *Ta*AA9A and *Nc*AA9F were stored in 50 mM Bis-Tris pH 6.5. Mn-dependent superoxide dismutase (Mn-SOD) from *E. coli* (Sigma-Aldrich, product number: S5639) was solubilized in Tris-HCl (10 mM, pH 8.0) and desalted (PD MidiTrap G-25, GE Healthcare) in the same buffer before use. Horseradish peroxidase (HRP, type II) (Sigma-Aldrich, product number: P8250) was solubilized in ultrapure Milli-Q treated water and filtered (Filtropur S, 0.2 μm PES, Sarstedt). All enzymes were stored at 4 °C.

### Standard photobiocatalytic LPMO reactions

Standard photobiocatalytic reactions were carried out in a cylindrical glass vial (1.1 mL) with a conical bottom (Thermo Scientific) with 500 μL reaction volume, unless otherwise specified. The light source (Lightningcure L9588, Hamamatsu) was equipped with a filter with a spectral distribution of 400–700 nm (L9588-03, Hamamatsu) and placed 1 cm above the liquid surface. Standard reactions contained *Sc*AA10C (0.5 μM), Avicel (10 g L$^{-1}$), and kraft lignin (0.9 g L$^{-1}$) in sodium phosphate buffer (50 mM; pH 7.0), unless otherwise specified. The reactions were incubated for 15 min in the dark at 40 °C under magnetic stirring prior to adding lignin and starting the reactions by turning on the light ($I = 10\%\ I_{max}$, equivalent to 16.8 W cm$^{-2}$). At regular intervals, 60 μL samples were removed from the reaction mixture and filtered using a 96-well filter plate (Millipore) and a vacuum manifold to stop the LPMO reaction. The filtered samples (35 μL) were stored at −20 °C prior to product quantification. A stock solution of recombinant, purified Cel6A from *Themobifida fusca* (*Tf*Cel6A)[73] was prepared in sodium phosphate buffer (50 mM; pH 6.0) and added to the filtrate to a final concentration of 2 μM, followed by incubation overnight at room temperature, to convert solubilized oxidized products to a mixture of C1-oxidized products with a degree of polymerization of 2 and 3 (GlcGlc1A and Glc$_2$Glc1A).

For measuring total oxidized products (i.e., both soluble and insoluble, as in Fig. 1), 50 μL samples were removed from the reaction, diluted with 50 μL H$_2$O and boiled for 15 min at 100 °C, cooled on ice, and stored at −20 °C prior to HPAEC-PAD analysis of oxidized products as described below. To prepare the samples for HPAEC-PAD analysis, 150 μL *Tf*Cel6A (5 μM final concentration) was added to 100 μL reaction suspension and the reaction was incubated in a thermomixer at 37 °C and 1200 rpm for 42 h to degrade all cellulosic material.

### Detection and quantification of LPMO products

Oxidized cello-oligosaccharides were analyzed by HPAEC-PAD performed with a Dionex ICS5000 system equipped with a CarboPac PA200 analytical column (3 × 250 mm) as previously described[54]. Chromatograms were recorded and analyzed using Chromeleon 7.0 software. Quantitative analysis of C1-oxidizing LPMO activity was based on quantification of cellobionic acid (GlcGlc1A) and cellotrionic acid (Glc$_2$Glc1A), which were obtained after treating reaction mixtures or reaction filtrates with *Tf*Cel6A, as described above. Standards of

GlcGlc1A and Glc$_2$Glc1A were prepared by treating cellobiose and cellotriose, both purchased from Megazyme, with cellobiose dehydrogenase[74].

Oxidized chito-oligosaccharides were qualitatively analyzed using an Agilent 1290 HPLC system with a HILIC column using UV-detection, as described elsewhere[75,76]. Chito-oligosaccharides with a degree of polymerization from 2 to 6 (Megazyme) were treated with a chito-oligosaccharide oxidase[77] to generate the corresponding oxidized chito-oligosaccharides[67], which were used as standards.

## H$_2$O$_2$ accumulation and consumption

The method for H$_2$O$_2$ detection was adapted from previously published methods[23,72] and modified as explained below. H$_2$O$_2$ accumulation in the light-exposed reactions containing lignin (0.9 or 9 g L$^{-1}$), LPMO (0, 75, or 500 nM), and Avicel (10 g L$^{-1}$) that are depicted in Fig. 1 was measured as follows: At given time points, 50 μL sample was withdrawn from the reaction and mixed with 50 μL H$_2$O before filtering as described above for LPMO reactions. 50 μL filtrate was recovered and diluted with water, after which 100 μL of diluted sample was mixed with 20 μL H$_2$O and 80 μL of a premix composed of HRP (0.4 μM) and AmplexRed (0.4 mM) in sodium phosphate buffer (0.4 M; pH 6.0). The H$_2$O$_2$ standard curve (0, 1, 2, 5, 10 μM) was prepared by mixing 80 μL of the same HRP/AmplexRed premix with 20 μL of an aqueous lignin solution to achieve approximately the same lignin concentration as for the reaction being measured, and lastly with 100 μL H$_2$O$_2$ solution (0, 2, 4, 10, 20 μM). All reaction mixtures were prepared in a non-transparent 96-well microtiter plate. The reaction mixtures were shaken for 30 s and incubated for 5 min at 30 °C prior to measuring fluorescence every 10 s for 2 min using 530/590 nm excitation/emission wavelengths in a Varioskan Lux plate reader (Thermo Fisher Scientific).

H$_2$O$_2$ consumption reactions were performed using the same conditions as the reactions for H$_2$O$_2$ production and were initiated by adding H$_2$O$_2$. Samples (50 μL) were withdrawn from the reaction at given time points (5, 10, 15, 40, 80, 120 min) and diluted with water prior to filtering the reaction mixture and measuring remaining H$_2$O$_2$, as described above.

## Transient state kinetics of LPMO reduction by lignin

We used the differences in intrinsic fluorescence between the Cu(II) and Cu(I) states of SmAA10A or ScAA10C to measure the kinetics of LPMO reduction by kraft lignin. Single-mixing experiments were carried out with a stopped-flow rapid spectrophotometer (SFM4000, BioLogic Science Instruments) coupled to a photomultiplier with an applied voltage of 600 V for detection. The excitation wavelength was set to 280 nm, and fluorescence was collected with a 340 nm bandpass filter. Single-mixing experiments were carried out by mixing LPMO-Cu(II) (5 μM final concentration after mixing, 50 mM sodium phosphate buffer, pH 7.0) with different concentrations of lignin (ranging from 1 to 100 μM final concentrations after mixing), in triplicates. All reagents were deoxygenated using a Schlenk line with N$_2$ flux and subsequently prepared in sealed syringes in an anaerobic chamber. The stopped-flow rapid spectrophotometer was flushed with a large excess of anaerobic buffer before coupling the sealed syringes and performing the experiments.

## Kinetics data analysis

The fluorescence data monitored with the stopped-flow was fitted to a single exponential function (y = a + b·e$^{-kobs·t}$) using the BioKine32 V4.74.2 software (BioLogic Science Instruments) to obtain the first order rate constant ($k_{1obs}$) for each lignin concentration. Plots of $k_{1obs}$ vs lignin concentration were fitted using linear least squares regression to obtain the apparent second order rate constant of the reduction step ($k_{1app}^{lignin}$) with SigmaPlot v14.0.

## NMR analyses

Kraft lignin (15 g L$^{-1}$) and organosolv lignin from birch or spruce (10 g L$^{-1}$) were incubated for 24 h in sodium phosphate buffer (50 mM, pH 7.0 for kraft lignin and pH 6.0 for organosolv lignin) at 40 °C under magnetic stirring, with or without exposure to visible light (I = 10% $I_{max}$, equivalent to 16.8 W cm$^{-2}$). For the incubations with organosolv lignin from birch or spruce, reactions were also performed in the presence of ScAA10C (500 nM) alone or ScAA10C (500 nM) in combination with Avicel (10 g L$^{-1}$), in the dark, to probe for putative LPMO-induced structural changes in the lignin. The reactions containing LPMO were performed as duplicates as opposed to the reactions treated with light or not in absence of LPMO, which were performed as four replicates. After 24 h, identical reactions were pooled and freeze-dried prior to NMR analyses.

Lyophilized organosolv lignin (20–40 mg) that had been incubated as described above was dissolved in 480 μL of deuterated dimethyl sulfoxide (DMSO-d$_6$ 99.96 atom % D Sigma-Aldrich) and transferred to a 5 mm LabScape Stream NMR tube (Bruker LabScape). For NMR analyses, all homo- and heteronuclear experiments were recorded on a Bruker AV-IIIHD 800 MHz spectrometer (Bruker BioSpin AG) equipped with a 5 mm cryogenic CP-TCI z-gradient probe. The spectra were recorded, processed, and analyzed using TopSpin 3.6pl7 and TopSpin 4.0.7 software (Bruker BioSpin AG).

For chemical shift assignments, the following one- and two-dimensional NMR experiments were recorded at 25 °C for both the birch and spruce lignin sample series: 1D carbon with power-gated decoupling and 30° flip angle (spectral width 220 ppm, spectral resolution 64k points, number of scans 4096, interscan delay 4 s), 1D proton with 30° flip angle (spectral width 14 ppm, spectral resolution 64k points, number of scans 16, interscan delay 1 s), 2D {$^1$H-$^{13}$C} heteronuclear single quantum coherence (HSQC) with multiplicity editing (spectral width C 200 ppm/ H 14 ppm, spectral resolution H 2k/ C 256k points, number of scans 32, interscan delay 2 s).

1D proton and carbon experiments were Fourier transformed using exponential windows function and line broadening of 0.3 Hz for proton and 5 Hz for carbon. Spectra were manually phase corrected with automatic baseline correction. HSQC experiments were Fourier transformed with the QSINE windows function (SSB = 2) in both dimensions, zero filling, linear prediction, and automatic baseline correction. All spectra were internally referenced to the residual DMSO signal (δ$_C$ 39.5 and δ$_H$ 2.50). Comparative analyses were only done for sets of reactions with similar lignin concentrations (i.e., those treated with LPMO containing ~20 mg lignin, and those treated with light, containing ~40 mg; the difference is due to sample availability). For presenting 1D spectra together, spectral intensities were scaled to the peak intensity at δ$_C$ ~28ppm and/or δ$_H$ 0.85 and 1.24, to compensate for differences in sample mass. Chemical moieties that changed, either in light-treated or LPMO-treated samples, were annotated based on comparison of chemical shift values with published literature values (see Figure captions for references).

$^1$H 1D NMR investigations of the kraft lignin were performed with a Bruker Avance III 400 MHz spectrometer equipped with a BBFO Plus double resonance probe head at 25 °C (Bruker BioSpin AG). 10–15 mg of lignin, treated as described above, was dissolved in 1500 μL of deuterated dimethyl sulfoxide (DMSO-d$_6$ 99.9 atom % D Sigma-Aldrich) and transferred to a 5 mm NMR tube. The spectra were acquired with 30° flip angle, spectral width 16 ppm, spectral resolution 64k points, number of scans 80, interscan delay 10 s. The spectra were recorded with TopSpin 3.64 (Bruker BioSpin AG). MestreNova software v14.1.1 was used for processing and analysis (Mestrelab research S.L.).

## Verification of superoxide dismutase activity

SOD activity was assessed using a published assay protocol[23,78]. In alkaline conditions, autooxidation of pyrogallol leads to formation

of $O_2^{\cdot-}$ which converts pyrogallol to purpurogallin, which absorbs strongly at 325 nm[78]. A stock solution of pyrogallol (15 mM in 10 mM HCl) was prepared in an aluminum foil wrapped tube and stored on ice and stock solutions of SOD were prepared in Tris-HCl (10 mM, pH 8.0) and kept on ice. All reactions were performed in 50 mM Tris-HCl pH 8.0 and were initiated by addition of pyrogallol (to 0.2 mM) immediately followed by addition of SOD (to 0, 10, 100, 1000 nM) and the absorbance at 325 nm was measured every 10 s for 3 min in a Hitachi U-1900 spectrophotometer. The inhibitory effect of SOD on pyrogallol autooxidation is shown in Supplementary Fig. 14.

### Reporting summary
Further information on research design is available in the Nature Portfolio Reporting Summary linked to this article.

## Data availability
The authors declare that all study data are included in the article and/ or the supplementary information. Data is also available from the corresponding author upon request. The UniProt IDs of the enzymes used in this study are Q9RJY2 (ScAA10C), O83009 (SmAA10A), G3XAP7 (TaAA9A), and Q1K4Q1 (NcAA9F).

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

## Acknowledgements

This work was supported by the Norwegian Research Council through grants 262853 (V.G.H.E), 257622 (S.J.H), 315385 (F.L.A), 268002 (V.G.H.E.), and 269408 (V.G.H.E), and by the European Commission through the ERC-SyG-2019 project CUBE with grant number 856446 (V.G.H.E.).

## Author contributions

E.G.K, S.J.H., and V.G.H.E designed the study. E.G.K., C.F.A., L.J.K., I.A.F., B.A., S.G.A., F.G., F.C., and F.L.A. performed the experiments. E.G.K., C.F.A., L.J.K., I.A.F., B.A., S.G.A., Y.S., D.E., F.G., F.C., S.J.H., F.L.A., and V.G.H.E. interpreted the data. E.G.K. and V.G.H.E. wrote the initial manuscript. All authors contributed to revising and writing the final version of the manuscript.

## Competing interests

The authors declare no competing interests.
