## [Peer Review File · Nature Communications]

Visible light-exposed lignin facilitates cellulose solubilization by lytic polysaccharide monooxygenasesReviewers' comments:

Reviewer #1 (Remarks to the Author):

This study investigated light-driven lignin for LPMO catalysis to facilitate cellulose solubilization and revealed the mechanism of H₂O₂-driven LPMO reaction. To prove their conclusions, the author used soluble kraft lignin and organosolv lignin as objects, and many biochemical experiments and lignin structure characterization were performed, but some data are still insufficient in my opinion. The evidence is not sufficient, and some experimental results are not credible. I have quite a few questions and comments that should be addressed.

1. The novelty of the article for *Nat Commun* is insufficient. There have been many reports on lignin-induced hydrogen peroxide production (Kim et al, <https://doi.org/10.1038/s44160-022-00035-2>). And hydrogen peroxide is the co-substrate of LPMO and could facilitate cellulose solubilization, which is generally known. So that exposing lignin to visible light facilitates cellulose solubilization by promoting formation of H₂O₂ that fuels LPMO catalysis is predictable.

2. In this paper, soluble kraft lignin (SKL) and organosolv lignin (OL) were used to study the hydrogen peroxide generation, LPMO activity and structural changes of lignin under light-induced conditions. For SKL, only the relationship between the hydrogen peroxide generation and LPMO activity was studied, but its structure of SKL was not characterized. And for OL, only the structure and LPMO activity was characterized, but hydrogen peroxide level was unknown. It is generally known that the structures and properties of SKL and OL are significantly different, and the structural changes of OL cannot explain the promotion mechanism of SKL. Similarly, the hydrogen peroxide level with SKL is also difficult to explain the promoting effect of OL on the LPMO reaction.

3. In this study, ¹H NMR, ¹³C NMR and 2D NMR were used to characterize the structure of OL to reveal the mechanism of production of hydrogen peroxide by light-induced lignin, but the NMR analysis was very rough and imperfect, lacking quantitative analysis data. ¹H NMR can only characterize the type and distribution of OH, ³¹P NMR should be used; ¹³C NMR lacks quantitative analysis; 2D NMR is a very effective semi-quantitative lignin structure characterization method. It is used to analyze changes in the relative abundance of lignin linkages and side chain structures. It is rarely analyzed by spectral stacking and should provide the raw integrated information; all NMR analyses lack information on signal attribution. The description of the NMR analysis method is also not detailed.

4. Lignin is a polymer with a complex structure, even if it is found that the ring-conjugated olefins in lignin are oxidized after light induction by NMR analysis, it is still necessary to use a series of methods to prove that the oxidation of this structure is related to the production of hydrogen peroxide. Specific lignin model compounds should be used to further identify the results, such as the use of structural analogs of cinnamyl alcohol.

5. Line 318, the author showed that lignin dimer cannot be induced to produce hydrogen peroxide by light, and it is proposed that it may be related to ring-conjugated double bonds in structures such as cinnamyl alcohol (Line 307-309). However, lignin dimer represents the main structure of lignin, and the abundance of cinnamyl alcohol and other structures in lignin is very low, which is difficult to explain the level of hydrogen peroxide production sufficient to drive peroxidase. In addition, the intensity of NMR signals using cinnamaldehyde end groups is easily disturbed by the process of sample preparation and analysis. It is difficult to compare the differences before and after light induction, and quantitative methods or model objects must be used to analyze.

6. As shown in Fig 6, the main signals of several spectra are not significantly different, and it is necessary to provide detailed integration data and relevant information of signal attribution.

7. Fig. 5 and Fig. 1a, it can be seen that the promotion effect of OL on LPMO activity is much lower than that of SKL. Whether this is due to the lower level of hydrogen peroxide production with OL, please provide evidence.

8. The soluble lignin used in this article is derived from Kraft pulping liquor, which only accounts for a very small part of the lignin in the liquor, while most of the Kraft lignin exists in the form of insoluble. The phenolic hydroxyl content of soluble kraft lignin is much higher than that of general lignin, and the molecular weight of soluble kraft lignin is also lower, similar to a polyphenolic polymer. But it is very common to use polyphenols to drive LPMO reactions. Therefore, a corresponding study should be carried out using the insoluble kraft lignin.

9. The lignin used by the authors is not representative. In addition to using insoluble kraft lignin, MWL and DHP should be used for further verification. The structure of MWL is closest to that of natural lignin.
10. Line 147, the author think that the affinity of SKL on hydrogen peroxide is lower than that of LPMO, but SKL rich in phenolic hydroxyl groups is a very effective ROS scavenger, Fig S2 also shows that in the absence and presence of LPMO, in the early reaction stage the scavenging ability of the hydrogen peroxide is not significant difference, indicating that lignin is the main scavenger of hydrogen peroxide. More sufficient evidence should be provided. For example, the reaction kinetics of SKL and LPMO on hydrogen peroxide.
11. Fig2 b, the author stated that avicel inhibits the lignin-induced hydrogen peroxide production, so the increase of avicel concentration leads to the decrease of LPMO activity, which should provide evidence of hydrogen peroxide production at different avicel concentrations.
12. Line 285, the pulping process did not lead to an increase in the abundance of C-O bonds.
13. Although Amplex Red reagent is used for the detection of LPMO enzyme activity, the method is affected by many factors, such as reducing agent, pH, heavy metal ions, etc. The author used Amplex Red assay to detect H₂O₂ accumulation and consumption, it may be inappropriate. Lignin is reducible, and its presence may affect the true value of hydrogen peroxide in the system. The author also emphasized this (Line 138-142). The method of H₂O₂ quantification using ABTS assay should be referred to.
14. Line 47-48, "LPMOs are mono-copper enzymes that catalyze oxidative cleavage of glycosidic bonds in cellulose and chitin", the statement is not comprehensive enough. Many studies have shown that LPMO can also participate in the oxidation of hemicellulose.
15. Line 96-97, the author indicated LPMOs oxidize lignin similar to what is observed for ligninolytic peroxidases and laccases. You really can't say that. Ligninolytic peroxidases and laccases can oxidize compounds with higher redox potentials and can also modify or break lignin linkages. Can LPMO also break lignin bonds? In addition, As far as I know, LPMO can usually only oxidize some phenolic compounds with lower redox potentials, such DMP, hydroquinone etc.
16. Line 107, "to assess the impact of light, we"
17. Line 108, "Avicel as substrate"
18. Line 111, "soluble kraft lignin", Please revise it throughout the manuscript. Kraft lignin is generally considered as insoluble kraft lignin.

Reviewer #2 (Remarks to the Author):

“Visible light-exposed lignin facilitates cellulose solubilization by lytic polysaccharide monoxygenases” by Kommedal et al describes work investigating the effect of visible light on the H₂O₂ producing ability of lignin and the effect that this has on LPMO activity. LPMOs are enzymes that have significant industrial value given their ability to improve the rate at which cellulose can be degraded which has partly driven interest in this area over the last 12 years. There is ongoing debate, however, as to the true co-substrate of these enzymes and whether they are peroxidases or oxidases. The Eijsink lab have done considerable work to demonstrate that H₂O₂ leads to much faster LPMO reaction kinetics, and this paper suggests another potential source of H₂O₂ to be harnessed by these enzymes. The authors have carefully demonstrated through a range of experiments that lignin, when exposed to light, can be a useful source of peroxide to drive the LPMO reaction. Using a chitin specific LPMO, they also confirm previous findings that LPMOs appear to be readily reduced by lignin, though it's odd that an alternative LPMO was required for this and the reason for this is not overly well justified (see comments below). The authors also perform some NMR analysis to try to understand the changes that take place in the lignin during the light dependent H₂O₂ production process which I have a couple of questions about though I have no expertise in this area whatsoever.

Overall, the paper is very well written, and the analysis appears to be sound for the most part and will be of interest to many working in the LPMO field. I find it a little difficult to place where the paper sits in terms of biological or industrial relevance. I will expand on this and some of my previous comments below, I hope that these will be useful to the authors and that in addressing them it will help improve what is already a very good manuscript.

COMMENTS

- 1.) In the paragraph starting at line 133 the authors seek to justify the fact that there are fewer apparent oxidised LPMO products produced compared to the amount of H₂O₂ produced as a result of lignin being exposed to light. I was not overly clear as to what the authors meant when they said “The levels of H₂O₂ measured in the absence of the LPMO are the net result of formation and degradation, both of which may be dependent on light, as has been shown for a different photoredox catalyst.” Could the authors please expand on what is meant here, as if there is light dependent H₂O₂ disproportionation of some sort then it is important to understand how this works? Did the authors do any experiments to check whether the levels of H₂O₂ drop over time after light exposure if the samples are returned to the dark to support this notion?
- 2.) At line 176 the authors note that at higher concentrations of Avicel the LPMO reaction appears less productive which they reason is due to absorbance of light resulting from the higher Avicel concentrations. There could be other explanations for this. Did the authors do any absorbance measurements or any controls to back up this statement? Seems like that would not be difficult to do and would firm up the conclusions.
- 3.) When attempting to measure LPMO reduction by lignin in stopped flow experiments the authors state at lines 239 to 241 that ScAA10C gave a weak fluorescence signal and quenching of fluorescence was observed in the presence of lignin and so they switched to a chitin dependent enzyme to enable them to perform the stopped flow experiments. Whilst I appreciate that this was a technical hurdle to overcome, is there any explanation for why the fluorescent signal should be weaker in ScAA10C as opposed to SmAA10A? And can the authors say anything about the rate of reduction in ScAA10C from the data they obtained? Supplementary figure 3 has normalised fluorescence data so it is hard to see but it appears that perhaps the rates of reduction are slower? I would expect at least some comment on this or plots of non-normalised fluorescence to better demonstrate the technical difficulties here.
- 4.) In the paragraph starting at line 247, the authors perform their experiments on lignin reduction with dialysed and native lignin in an attempt to remove low molecular weight species from the lignin. Did the authors do anything to check that this had removed some low molecular weight species? I

appreciate they state this is normal practice with manganese peroxidases for instance to remove any Mn^{2+} knocking around, but low molecular weight lignin species are likely harder to remove through dialysis so I would expect the authors to have done something to check that there genuinely was an effect on low molecular weight species from this treatment.

5.) In the NMR analysis in Figure 6 and Supplementary figure 6, the figure legends state that the samples containing LPMO had about half the amount of lignin compared to samples without the LPMOs present and so the signals were essentially scaled. I question whether this is a fair thing to do without more significant justification in the main text. The reason for the discrepancies in concentrations should really be explained at the very least. The background noise is clearly higher in the LPMO containing samples so how do the authors know that any similar signals are not simply lost in the noise in the LPMO containing samples as there are clearly some small signals at the same chemical shift in the dark, non-LPMO sample?

6.) The final results paragraph relating to whether the H_2O_2 could have been generated as a result of H_2O reduction as opposed to O_2 reduction appears out of place to me. This may have been a late addition to the paper or done in response to a previous submission to another journal, but I think it would sit better earlier on following the initial demonstration of H_2O_2 production as a result of light treatment and the demonstration of LPMO activity under such conditions. In addition, none of the MALDI data that are described is shown either in the main text or the SI, this should really be included.

7.) Generally speaking, it is hard to see either the industrial or biological relevance of the research as presented. There is no doubt that the results support that light induced H_2O_2 production can occur from lignin (which I believe is a known phenomenon anyway) but how likely is it that this occurs in nature to a significant extent when LPMOs are present? Fungi are the most heavy users of LPMOs but they typically grow in dark, damp places as far as I am aware and so light is unlikely to have significant impact. How do the levels of light used in this research compare to the light intensities that would be expected in the habitats of such fungi or indeed soil dwelling bacteria? I appreciate that the authors note the caveats as relate to the potential industrial applications of this research but I think the likely biological significance of the findings should be discussed in greater depth as well.

MINOR CORRECTIONS

Line 89 – explaining should be “explain”

Line 259 – I think it is more normal practice to write “ $3.7 \cdot 10^3 M.s^{-1}$ and $2.9 \cdot 10^3 M.s^{-1}$ ” as “ $3.7 \times 10^3 M.s^{-1}$ and $2.9 \times 10^3 M.s^{-1}$ ”

Line 428 – “a cheap and abundant...” should be “a cheap and abundant...”

Lines 430 to 436 appear to be missing some references.

Lines 467 to 469 – Is there a reference to the effect of ploughing and how this fits in here?

Revision of

Visible light-exposed lignin facilitates cellulose solubilization by lytic polysaccharide monoxygenases" (NCOMMS-22-24127-T)

Rebuttal letter that addresses all referee comments in a point-by-point manner.

The text below contains all referee comments.

Line numbers refer to the revised manuscript, in which changed parts are marked in red.

Please note that two Supplementary Figures have been added to become Figs. S3 and S12. The numbers of the other Supplementary Figures have been adjusted accordingly. Figures 4 and 6, and Supplementary Figure 6 (now Supplementary Figure 7), have been revised.

REVIEWER #1

COMMENT: This study investigated light-driven lignin for LPMO catalysis to facilitates cellulose solubilization and revealed the mechanism of H₂O₂-driven LPMO reaction. To prove their conclusions, the author use soluble kraft lignin and organosolv lignin as objects, and many biochemical experiments and lignin structure characterization was performed, but some data are still insufficient in my opinion. The evidence is not sufficient, and some experimental results are not credible. I have quite a few questions and comments that should be addressed.

RESPONSE: We largely disagree with this reviewer, as outlined in more detail below. The core problems raised are novelty (see below), which we think is severely misjudged, and lack of sufficient experiments to back up our conclusions. As to the latter, we agree that "more could have been done" (as is always the case in scientific research), but we disagree that more **MUST** be done to support the most important of our conclusions or warrant publication.

Instead of focusing on what has been done, this reviewer focuses on all that could have been done in addition. This is not okay, but may perhaps be explained by this reviewer's (in our view incorrect) perception of a lack of novelty (see next point). The comments on the lignin analysis and quantification, which also the Editor highlighted in the response to us, are valid, but there are serious technical limitations which seem to be overlooked. As to the lignin characterization work (NMR), there is a valid rationale behind our choices, as outlined below.

COMMENT: 1. The novelty of the article for Nat Commun is insufficient. There have been many reports on lignin-induced hydrogen peroxide production (Kim et al, <https://doi.org/10.1038/s44160-022-00035-2>). And hydrogen peroxide is the co-substrate of LPMO and could facilitates cellulose solubilization, which is generally known. So that exposing lignin to visible light facilitates cellulose solubilization by promoting formation of H₂O₂ that fuels LPMO catalysis is predictable.

RESPONSE: It is indeed true that H₂O₂ production upon irradiating lignin with light has been known for a while. Furthermore, there is an excellent and exciting 2022 paper (Kim et al., 2022, Nature Synthesis; pointed at in the reviewer's comment and cited by us) showing that one can drive peroxygenase reactions with H₂O₂ produced by light-irradiated lignin. Indeed, we were to some extent "scooped" by this 2022 Kim paper (March 2022), which, as far as we know, is the only paper showing something similar to what we show. We would respectfully claim that novelty, in terms of

driving a peroxygenase reaction with irradiated lignin, a very important phenomenon from a sustainability perspective, is still quite high. However, there is more: There is huge novelty for the hot fields of LPMO catalysis and biomass conversion, which the reviewer does not seem to appreciate at all. The reviewer's claim that H₂O₂-driven LPMO activity is generally accepted in the field is incorrect, and the implications of this paradigm are only just beginning to emerge (as, e.g., in the present manuscript). Looking at the issue from an "LPMO perspective", we would argue:

- It is wrong to claim that the results are "predictable". They are for me, and, apparently, for the reviewer, but not for the field. Note the comment by reviewer #2: "There is ongoing debate, however, as to the true co-substrate of these enzymes and whether they are peroxidases or oxidases."
- Even if the results are "predictable", it is still a major achievement to observe and understand the processes. LPMO reactions are notoriously difficult to control and understand. We show how to do that, using a light fueled lignin-based reaction system. This is unprecedented.
- Most importantly, and perhaps somewhat under communicated in the present manuscript, lignin is abundantly present in the co-polymeric substrates (lignocellulosic biomass) on which LPMOs work, both in nature and industrial biorefining. So, the photocatalyst is present in the LPMO substrates themselves! This is highly interesting as such, and also means that LPMO reactions with lignocellulose, carried out in many laboratories and industries all over the world, are light sensitive. Until now, this has not been considered at all, neither in academia, nor in industry. Simply making the "LPMO field" aware of the light-sensitivity of lignin will have major impact.
- For one of the lignins, we show what happens to the lignin in LPMO reactions and we show that light oxidation and LPMO oxidation have different effects on the lignin. This has not been done before.

Regarding our third argument, we have made the following change to the manuscript:

- At the end of the fifth paragraph of the Introduction, we have added the following sentence: "Of note, possible effects of light may also be relevant for reactor design in industrial biorefining of lignocellulosic biomass, since pretreated feedstocks that are subjected to enzymatic saccharification with LPMO-containing cellulolytic enzyme cocktails usually contain large amounts of lignin."

The (potential) industrial relevance of our work is also addressed in the (revised) Discussion section; please see our response to comment # 7 by Reviewer #2.

COMMENT: 2. In this paper, soluble kraft lignin (SKL) and organosolv lignin (OL) were used to study the hydrogen peroxide generation, LPMO activity and structural changes of lignin under light-induced conditions. For SKL, only the relationship between the hydrogen peroxide generation and LPMO activity was studied, but its structure of SKL was not characterized. and for OL, only the structure and LPMO activity was characterized, but hydrogen peroxide level was unknown. It is generally known that the structures and properties of SKL and OL are significantly different, and the structural changes of OL cannot explain the promotion mechanism of SKL. Similarly, the hydrogen peroxide level with SKL is also difficult to explain the promoting effect of OL on the LPMO reaction.

RESPONSE: We agree that some of these issues could have been addressed, but we do not agree that this is needed to substantiate our conclusions. This is just asking for "more". We used two types of lignin to show that our findings apply to multiple lignin types (which adds value to the work). Of note, Kim et al., used yet other types of lignin, in their 2022 study of light-promoted peroxygenase activity. Of course, there may be variations between the various lignin types, but the overall trend is clear and we do not generalize in terms of the structural changes that may occur in these different lignins. We would also like to point out that full structural characterization of the lignins used, while

interesting, is outside the scope of this paper. We have, for good reasons (see also below) focused on what is essential.

COMMENT: 3. In this study, ¹H NMR, ¹³C NMR and 2D NMR were used to characterize the structure of OL to reveal the mechanism of production of hydrogen peroxide by light-induced lignin, but the NMR analysis was very rough and imperfect, lacking quantitative analysis data. ¹H NMR can only characterize the type and distribution of OH, ³¹P NMR should be used; ¹³C NMR lacks quantitative analysis; 2D NMR is a very effective semi-quantitative lignin structure characterization method. It is used to analyze changes in the relative abundance of lignin linkages and side chain structures. It is rarely analyzed by spectral stacking and should provide the raw integrated information; all NMR analyses lack information on signal attribution. The description of the NMR analysis method is also not detailed.

RESPONSE: We acknowledge that additional analysis of the lignins could be interesting, but believe that this is out of scope or, in any case, not essential. Furthermore, there are technical and scientific hurdles that the reviewer perhaps does not appreciate. The aim of the included NMR analyses was to investigate qualitative structural changes in lignin treated with light/LPMO, specifically looking for alpha-carbon oxidation. We agree that ³¹P NMR can be used to distinguish different types of -OH groups and using ³¹P NMR for a more thorough characterization would be a good suggestion for future work but is beyond the scope of the present study. We fail to see the relevance of attempting quantitative analysis due to both the varying and limited solubility and the complexity (heterogeneity) of native lignin structures. Solution-state NMR is limited to detect only those compounds that are dissolved within the solvent system, whereas insoluble lignin clearly affects the LPMO reaction too. Therefore, truly quantitative experiments are hardly possible. Q-HSQC/2D-HSQC spectra will only reflect lignin in solution at the given condition, are less sensitive, and are somewhat sensitive to scalar coupling and relaxation. Therefore, we used a qualitative approach to characterize the changes in the lignin, using normal HSQC spectra (providing better sensitivity compared to quantitative methods). We think that this is valid and sufficient. A more quantitative approach would not change the conclusions and main message of the paper, and would introduce ambiguities with accompanying side-track discussions and speculation, as alluded to above. Specifically, neither the HSQC nor the 1D carbon pulse sequence is quantitative, so raw integrated information is not applicable. Information on signal attribution is provided; see relevant references in the captions of Figure 6, and Supplementary Figures 7-9 (previously Supplementary Figures 6-8).

We fully agree with the reviewer that the original description of the NMR methods was insufficient. For further clarification and to illustrate the nature and robustness of our data, we have made the following changes:

- In the first sentence of the first paragraph describing the NMR studies, the word “qualitatively” has been added to a sentence that now reads: “NMR spectroscopy was used to qualitatively investigate light-induced and LPMO-induced changes in the lignin structures directly.”
- Figure 6 (NMR analysis of spruce organosolv lignin) and its legend have been reorganized to more convincingly show the effects (or absence thereof) of light and LPMO action on lignin structure. The changes in the legend are relatively minor. The Figure has now six (a-f), instead of four (a-d) panels and appropriate minor changes have been made throughout.
- Similar changes were made for Supplementary Figure 6 (now Supplementary Figure S7; NMR analysis of birch organosolv lignin).
- We have added a “cautionary” paragraph to the Discussion section, addressing, among other things, the challenges associated with obtaining more quantitative data. Please see our response to comment #5 by this reviewer, below.

- A paragraph providing a detailed description of how NMR spectra were processed and interpreted has been added to the Materials and Methods section (we apologize that such a paragraph was lacking in the original manuscript). This new paragraph reads:
“1D proton and carbon experiments were Fourier transformed using exponential windows function and line broadening of 0.3 Hz for proton and 5 Hz for carbon. Spectra were manually phase corrected with automatic baseline correction. HSQC experiments were Fourier transformed with the QSINE windows function (SSB=2) in both dimensions, zero filling, linear prediction, and automatic baseline correction. All spectra were internally referenced to the residual DMSO signal (δ_C 39.5 and δ_H 2.50). Comparative analyses were only done for sets of reactions with similar lignin concentrations (i.e., those treated with LPMO containing ~20 mg lignin, and those treated with light, containing ~40 mg; the difference is due to sample availability). For presenting 1D spectra together, spectral intensities were scaled to the peak intensity at δ_C ~28ppm and/or δ_H 0.85 and 1.24, to compensate for differences in sample mass. Chemical moieties that changed, either in light-treated or LPMO-treated samples, were annotated based on comparison of chemical shift values with published literature values (see Figure captions for references).”

COMMENT: 4. Lignin is a polymer with a complex structure, even if it is found that the ring-conjugated olefins in lignin are oxidized after light induction by NMR analysis, it is still necessary to use a series of methods to prove that the oxidation of this structure is related to the production of hydrogen peroxide. Specific lignin model compounds should be used to further identify the results, such as the use of structural analogs of cinnamyl alcohol.

RESPONSE: We fail to see why this is “necessary” to support the conclusions of our study. Furthermore, we would argue that the use of model compounds bears little relevance for what we are showing and discussing because these model compounds differ from the much more complex, less soluble, and in the case of organosolv lignin, more natural lignins used in this study.

COMMENT: 5. Line 318, the author showed that lignin dimer cannot be induced to produce hydrogen peroxide by light, and it is proposed that it may be related to Ring-conjugated double bonds in structures such as cinnamyl alcohol (Line 307-309). However, Lignin dimer represents the main structure of lignin, and the abundance of cinnamyl alcohol and other structures in lignin is very low, which is difficult to explain the level of hydrogen peroxide production sufficient to drive peroxidase. In addition, the intensity of NMR signals using cinnamaldehyde end groups is easily disturbed by the process of sample preparation and analysis. It is difficult to compare the differences before and after light induction, and quantitative methods or model objects must be used to analyze.

RESPONSE: We acknowledge the technical challenges pointed at by the reviewer, which is exactly why we are careful and not quantitative in our conclusions (see also our response to comment #3 by this reviewer). In fact, the reviewer makes a quantitative claim in the comment (“the abundance of cinnamyl alcohol and other structures in lignin is very low, which is difficult to explain the level of hydrogen peroxide production sufficient to drive peroxidase”) for which there is no basis (since there are no quantitative data to compare and since such data are almost impossible to generate). Besides, the level of cinnamyl alcohols is not that low (e.g., Martín-Sampedro et al., 2019, Int J Biol Macromol 126:18-29) and may as such be high enough to drive the reaction. Most importantly though, we can only describe what we see, which is an increase in cinnamaldehydes and a decrease in stilbenes. This does not mean that there cannot be other changes in the lignin.

After much careful optimization of our experiments, we were able to reliably detect, by NMR, effects of light on lignin. We do not agree that including “quantitative methods or model objects must be used to analyze”, for reasons discussed above. One cannot compare these model compounds with the lignins used in our study. Of note, we did use one model compound (the lignin dimer referred to by the reviewer), which did not work in our hands, in contrast to what has been reported by Kim et al., in 2022. The fact our results deviate somewhat from those of Kim et al. (model compounds) should not be used against us.

We acknowledge that we could have presented things in a better way and we hope that the changes made in response to comment #3 by this reviewer (see above) improve the situation.

We agree that the paper would benefit from a more clear and critical discussion of the quantitative limitations of these NMR analyses. This would also assist the non-specialist to interpret the results with the adequate amount of caution. Therefore, we have added a cautionary paragraph to the Discussion section, which reads as follows:

“Importantly, while the structural studies of lignin show effects of both irradiation and LPMO action and clearly point at the chemical processes involved, further studies are needed to fully unravel structural changes in lignin. We used the highest practical sample concentrations in the NMR analyses, to maximize sensitivity. The complexity and heterogeneity of the lignin structures requires high sensitivity, while achieving complete dissolution of samples is challenging. It is likely that the structural changes in lignin observed in this study only provide part of the picture, due to low signal-to-noise ratios, particularly for the 1D carbon spectra. Of note, the apparent lack of an effect of LPMO treatment on the 1D carbon spectra of lignin (Figure 6 & Supplementary Figure 7) could to some extent be due to the lower signal-to-noise ratio in these spectra (compared to the spectra obtained in the experiments with light). Thus, we cannot fully exclude that LPMO action also leads to lignin oxidations similar to those occurring upon treatment with light. Further in-depth studies of treated and untreated lignin are needed to unravel the full impact of light and LPMO action of lignin. Such studies may eventually allow the determination of quantitative correlations between the degree of lignin oxidation, the amount of hydrogen peroxide produced and LPMO activity. Of note, revealing such correlations would require accurate quantitative detection of all LPMO products and hydrogen peroxide levels under relevant conditions, which is challenging for reactions with lignin.”

COMMENT: 6. As shown in Fig6, the main signals of several spectra are not significantly different, and it is necessary to provide detailed integration data and relevant information of signal attribution.

RESPONSE: This is a peculiar comment since several of the spectra are “expected” to not be different. The differences that do occur illustrate the effect of light on lignin (panel a versus b; now panels a,b,f) and the lack of effect of the LPMO reaction on lignin (panel c versus d, the clear changes all relate to cellulose degradation and not to lignin modification; now panels c, d, e). These observable qualitative changes cannot be disputed (and we do not understand why the reviewer does so; the signals are clear). The extent of their formal statistical significance is something that should be determined with the relevant quantitative measure and statistical analyses and we believe that the present results provide a good starting point for such further investigations. We would also like to stress, again, that none of our conclusions are quantitative and that this is for good reasons. Even if we had quantitative data on lignin oxidation, it would not be possible to link this, quantitatively, to the availability of H₂O₂, or catalytic activity, in LPMO reactions. The dose-response studies depicted in Fig. 2 and the control experiments depicted in Fig. 3 very clearly show that the LPMO reaction is driven by the combination of lignin and light and that H₂O₂ production is key.

We have made multiple changes to the NMR parts of the manuscript that should improve clarity, as described in our response to comments #3 and #5 by this reviewer and comment #5 by Reviewer #2. Of note, the carbon spectra are simply not quantitative so it would be incorrect to use integration data, as should be clear from the now improved Methods section.

We provide references for signal attribution. We could provide the exact chemical shifts for annotated signals if the Editor or Reviewer insist, but we feel that it is better not to do so, as we are referring to general changes in chemical moieties, some of which we likely do not see in our data (see comment #5 by this Reviewer, above).

Figure 6 has been changed to more clearly highlight the changes (or lack thereof) caused by exposure to light or to the LPMO.

COMMENT: 7. Fig. 5 and Fig. 1a, it can be seen that the promotion effect of OL on LPMO activity is much lower than that of SKL. Whether this is due to the lower level of hydrogen peroxide production with OL, please provide evidence.

RESPONSE: We understand this request, but, at the same time, we find it not reasonable. First of all, the two experiments really cannot be compared, quantitatively (different lignin concentrations, soluble vs insoluble lignin, pH). Secondly, in the preceding part of the manuscript, we have very carefully shown, with all kinds of controls, that the light/lignin mediated stimulation of LPMO activity relates to H₂O₂ production. So, is it reasonable to show this for another type of lignin? (What else could it be?). In fact, there is ample data in the LPMO literature that would “turn this critique around”: LPMO activity is a good way to detect H₂O₂ production in complex reaction systems.

COMMENT: 8. The soluble lignin used in this article is derived from Kraft pulping liquor, which only accounts for a very small part of the lignin in the liquor, while most of the Kraft lignin exists in the form of insoluble. The phenolic hydroxyl content of Soluble kraft lignin is much higher than that of general lignin, and the molecular weight of Soluble kraft lignin is also lower, similar to a polyphenolic polymer. But it is very common to use polyphenols to drive LPMO reactions. Therefore, a corresponding study should be carried out using the insoluble kraft lignin.

RESPONSE: These are valid points except for the crucial conclusion: “Therefore, a corresponding study should be carried out using the insoluble kraft lignin.” Why? What essential information would this add? Why is an in-depth analysis of insoluble Kraft lignin important? Besides, we think that the work with organosolv lignin provides quite some of the desired information and this lignin is closer to “natural” lignin.

COMMENT: 9. The lignin used by the authors is not representative. In addition to using insoluble kraft lignin, MWL and DHP should be used for further verification. The structure of MWL is closest to that of natural lignin.

RESPONSE: With all due respect, this is not really fair (see also previous point). One can always do more. And how does one judge what is sufficiently representative? How much would the use of additional lignin types add to this paper? Of note, from Kim et al., 2022, it is already known that other lignin types, including DHP generate H₂O₂ when exposed to light. The requested additional experiments are not at all needed to substantiate our findings.

COMMENT: 10. Line 147, the author think that the affinity of SKL on hydrogen peroxide is lower than that of LPMO, but SKL rich in phenolic hydroxyl groups is a very effective ROS scavenger, Fig S2 also shows that in the absence and presence of LPMO, in the early reaction stage the scavenging ability of the hydrogen peroxide is not significant difference, indicating that lignin is the main

scavenger of hydrogen peroxide. More sufficient evidence should be provided. For example, the reaction kinetics of SKL and LPMO on hydrogen peroxide.

RESPONSE: This comment relates to a careful speculation phrased as follows: “Since LPMOs in presence of substrate have high affinity for H₂O₂ (K_m values in the low micromolar range;^{9,27,40} it is conceivable that the LPMO peroxygenase reaction outcompetes consumption of H₂O₂ through reactions with lignin, which would explain the discrepancy between apparent H₂O₂ measured and LPMO product levels.” This speculation is not central for the overall conclusions of the paper and would be supported by most people working with LPMOs. It is generally thought, and has been shown, that LPMOs react very fast with H₂O₂ if a suitable substrate is present and will likely outcompete most abiotic reactions involving H₂O₂. We agree with the reviewer’s formal description of Fig. S2, but do not agree with the subsequent conclusion that “that lignin is the main scavenger of hydrogen peroxide.” This may very well be so in the very initial phase of reactions (i.e., the first 30 minutes), but Figure S2 clearly shows that LPMO catalysis is a much more efficient H₂O₂ consumer than abiotic reactions with lignin in the time scale typically used in the LPMO reactions in the manuscript. Those time scales are in the order of 6 hours, whereas Fig. S2 covers only two hours.

To clarify this issue, we have added the following sentence to the legend of Fig. S2: “The curves show that after an initial phase of equally fast H₂O₂ consumption lasting some 30 min, the reaction with the LPMO activity leads to faster H₂O₂ consumption in the later phase of the reaction (note that LPMO reactions reported in the manuscript typically lasted 6 hours).”

COMMENT: 11. Fig2 b, the author stated that avicel inhibits the lignin-induced hydrogen peroxide production, so the increase of avicel concentration leads to the decrease of LPMO activity, which should provide evidence of hydrogen peroxide production at different avicel concentrations.

RESPONSE: If we understand correctly, the reviewer would like us to document that the presence of Avicel reduces hydrogen peroxide production. Light attenuation by varying concentrations of Avicel has been reported previously in Fig. S3 of Blossom et al. 2020 (*ACS Sustain. Chem. Eng.* **8**, 9301–9310, 2020), but this paper does not report on effects on hydrogen peroxide production. We cite this paper, but this is, as the reviewer points out, not sufficient. Therefore, we have done additional experiments that support the suggestion that Avicel inhibits light-lignin-induced hydrogen peroxide production. They appear as new Figure S3. The following text has been added to the Result section (Lines 181-183):

“Control reactions without enzyme showed that, indeed, the production of H₂O₂ in light-exposed reactions with a fixed amount of lignin is inversely correlated with the Avicel concentration (Supplementary Figure 3).”

Of note, the other reviewer also commented on this.

COMMENT: 12. Line 285, the pulping process did not lead to an increase in the abundance of C-O bonds.

RESPONSE: We do not understand this comment since we do not state that pulping leads to an increase in the abundance of C-O bonds. We say that the pulped material contains other, more recalcitrant C-O bonds rather than the less recalcitrant beta-O-4 bonds, which is in line with the information provided in the two cited papers. (The text reads: “This process generates a modified and condensed lignin structure with an increase in phenolic groups and recalcitrant C-C and C-O bonds, and a reduced number of less recalcitrant β-O-4 bonds, compared to native lignin^{46,47}.”)

COMMENT: 13. Although Amplex Red reagent is used for the detection of LPMO enzyme activity, the method is affected by many factors, such as reducing agent, pH, heavy metal ions, etc. The author used Amplex Red assay to detect H₂O₂ accumulation and consumption, it may be inappropriate. Lignin is reducible, and its presence may affect the true value of hydrogen peroxide in the system. The author also emphasized this (Line 138-142). The method of H₂O₂ quantification using ABTS assay should be referred to.

RESPONSE: We are fully aware of the pitfalls of the Amplex Red assay and these have all been accounted for, as outlined in the Materials and Methods section. Interestingly, a review paper about these pitfalls, written by our team, has just been accepted for publication in *Methods in Enzymology*. In reactions with lignin, in our hands, Amplex Red works better than ABTS. In any case, we ask, what would another H₂O₂ assay add? Our paper is, for good technical reasons, semi-quantitative. The trends are clear; the absolute numbers, if they could be obtained, will not add much.

COMMENT: 14. Line 47-48, "LPMOs are mono-copper enzymes that catalyze oxidative cleavage of glycosidic bonds in cellulose and chitin", the statement is not comprehensive enough. Many studies have shown that LPMO can also participate in the oxidation of hemicellulose.

RESPONSE: This is correct and we have changed the text accordingly. The revised text reads: "LPMOs are mono-copper enzymes^{4,5} that catalyze oxidative cleavage of glycosidic bonds in insoluble polysaccharides such as cellulose^{5,6} and chitin³, as well as in certain hemicelluloses^{8,9}."

COMMENT: 15. Line 96-97, the author indicated LPMOs oxidize lignin similar to what is observed for ligninolytic peroxidases and laccases. You really can't say that. Ligninolytic peroxidases and laccases can oxidize compounds with higher redox potentials and can also modify or break lignin linkages. Can LPMO also break lignin bonds? In addition, As far as I know, LPMO can usually only oxidize some phenolic compounds with lower redox potentials, such DMP, hydroquinone etc.

RESPONSE: This is correct. Our apologies for this mistake, which takes things "too far". We do not know the eventual consequence of LPMO-catalyzed oxidation of lignin on lignin integrity and it is thus not appropriate to make the comparison with peroxidases and laccases, as we did. The words "similar to what is observed for ligninolytic peroxidases and laccases" have now been deleted.

COMMENT: 16. Line 107, "to assess the impact of light, we"

RESPONSE: A comma has been added, as suggested.

COMMENT: 17. Line 108, "Avicel as substrate"

RESPONSE: The words "as substrate" have been added, as suggested.

COMMENT: 18. Line 111, "soluble kraft lignin", Please revise it throughout the manuscript. Kraft lignin is generally considered as insoluble kraft lignin.

RESPONSE: We do not understand this comment and/or we respectfully disagree. We specify that the kraft lignin we are working with is water soluble. There is no water insoluble fraction in this

lignin. We have purchased it from Sigma and we provide the product number in the Materials and Methods section.

REVIEWER #2

COMMENT: “Visible light-exposed lignin facilitates cellulose solubilization by lytic polysaccharide monoxygenases” by Kommedal et al describes work investigating the effect of visible light on the H₂O₂ producing ability of lignin and the effect that this has on LPMO activity. LPMOs are enzymes that have significant industrial value given their ability to improve the rate at which cellulose can be degraded which has partly driven interest in this area over the last 12 years. There is ongoing debate, however, as to the true co-substrate of these enzymes and whether they are peroxidases or oxidases. The Eijsink lab have done considerable work to demonstrate that H₂O₂ leads to much faster LPMO reaction kinetics, and this paper suggests another potential source of H₂O₂ to be harnessed by these enzymes. The authors have carefully demonstrated through a range of experiments that lignin, when exposed to light, can be a useful source of peroxide to drive the LPMO reaction. Using a chitin specific LPMO, they also confirm previous findings that LPMOs appear to be readily reduced by lignin, though it’s odd that an alternative LPMO was required for this and the reason for this is not overly well justified (see comments below). The authors also perform some NMR analysis to try to understand the changes that take place in the lignin during the light dependent H₂O₂ production process which I have a couple of questions about though I have no expertise in this area whatsoever.

Overall, the paper is very well written, and the analysis appears to be sound for the most part and will be of interest to many working in the LPMO field. I find it a little difficult to place where the paper sits in terms of biological or industrial relevance. I will expand on this and some of my previous comments below, I hope that these will be useful to the authors and that in addressing them it will help improve what is already a very good manuscript.

RESPONSE: We are happy that the reviewer found our manuscript interesting, “very well written” and “very good”. We are grateful for the insightful comments that helped improving the manuscript.

COMMENT: 1.) In the paragraph starting at line 133 the authors seek to justify the fact that there are fewer apparent oxidised LPMO products produced compared to the amount of H₂O₂ produced as a result of lignin being exposed to light. I was not overly clear as to what the authors meant when they said “The levels of H₂O₂ measured in the absence of the LPMO are the net result of formation and degradation, both of which may be dependent on light, as has been shown for a different photoredox catalyst.” Could the authors please expand on what is meant here, as if there is light dependent H₂O₂ disproportionation of some sort then it is important to understand how this works? Did the authors do any experiments to check whether the levels of H₂O₂ drop over time after light exposure if the samples are returned to the dark to support this notion?

RESPONSE: There seems to be a misunderstanding here, since the reviewer writes “there are fewer apparent oxidised LPMO products produced compared to the amount of H₂O₂ produced as a result of lignin being exposed to light”. It is actually the other way around. In this paragraph we seek explanations for the difference and we believe that there is at least one that is quite plausible, as phrased in lines 145-153, which appear at the end of our answer to this comment. This explanation was commented upon by Reviewer #1, Comment #10, and we refer to our response to this comment

for further details. Reviewer #1 did not disagree with our focus on abiotic reactions. We did not consider light dependent H₂O₂ disproportionation.

In accordance with the reviewer's comment, we have changed the text for clarification (lines 145-149):

Old text: "Another explanation lies in the abiotic consumption of H₂O₂. The levels of H₂O₂ measured in the absence of the LPMO are the net result of formation and degradation, both of which may be dependent on light, as has been shown for a different photoredox catalyst⁴², and on the presence redox-active moieties in the lignin itself⁴³."

New text: "Another explanation lies in the abiotic consumption of H₂O₂ due to reactions with lignin⁴⁴. The levels of H₂O₂ measured in the absence of the LPMO are the net result of formation (i.e., oxidation of lignin by O₂) and degradation (i.e., oxidation of lignin by H₂O₂), both of which may be dependent on light, as has been shown for a different photoredox catalyst⁴⁵."

This revised text is followed by a crucial sentence (not changed during the revision) explaining why the situation is different in reactions with the LPMO:

"Since LPMOs in presence of substrate have high affinity for H₂O₂ (K_m values in the low micromolar range)^{11,29,43} it is conceivable that the LPMO peroxygenase reaction outcompetes consumption of H₂O₂ through reactions with lignin, which would explain the discrepancy between apparent H₂O₂ measured and LPMO product levels."

COMMENT: 2.) At line 176 the authors note that at higher concentrations of Avicel the LPMO reaction appears less productive which they reason is due to absorbance of light resulting from the higher Avicel concentrations. There could be other explanations for this. Did the authors do any absorbance measurements or any controls to back up this statement? Seems like that would not be difficult to do and would firm up the conclusions.

RESPONSE: We have now done a key control experiment showing that, indeed, Avicel inhibits light-lignin-induced hydrogen peroxide production, which will reduce LPMO activity, as outlined in more detail in our response to comment #11 by reviewer #1. The new results are shown in Figure S3 and the following text has been added to the Result section (Lines 181-183):

"Control reactions without enzyme showed that, indeed, the production of H₂O₂ in light-exposed reactions with a fixed amount of lignin is inversely correlated with the Avicel concentration (Supplementary Figure 3)."

COMMENT: 3.) When attempting to measure LPMO reduction by lignin in stopped flow experiments the authors state at lines 239 to 241 that ScAA10C gave a weak fluorescence signal and quenching of fluorescence was observed in the presence of lignin and so they switched to a chitin dependent enzyme to enable them to perform the stopped flow experiments. Whilst I appreciate that this was a technical hurdle to overcome, is there any explanation for why the fluorescent signal should be weaker in ScAA10C as opposed to SmAA10A? And can the authors say anything about the rate of reduction in ScAA10C from the data they obtained? Supplementary figure 3 has normalised fluorescence data so it is hard to see but it appears that perhaps the rates of reduction are slower? I would expect at least some comment on this or plots of non-normalised fluorescence to better demonstrate the technical difficulties here.

RESPONSE: The chitin-active LPMO, *SmAA10A*, gives a much stronger change in fluorescence signal upon reduction, which is particularly important in reactions containing spectroscopically active compounds such as lignin. This means that with increasing concentrations of lignin, the fluorescence quenching that prevents obtaining reliable data for *ScAA10C* does not prevent us from obtaining such data for *SmAA10A*. Thus, a full kinetics experiment with lignin could only be done with *SmAA10A*.

The different fluorescence properties depending on the redox state of the LPMO are not fully understood, but the stronger reduction dependency of the signal for *SmAA10A* is likely due the following: (1) *SmAA10A* has more tryptophans near the copper ion; this improves signal strength; (2) *ScAA10C* has an additional domain that contains tryptophans which give a fluorescence signal that is not affected by the redox state of the copper. This high “background” signal complicates quantitative stopped-flow studies with *ScAA10C*; the sensitivity is low. We have added this information to the legend of Fig. S3 (see below). In the main text, line 246-247, we have replaced “(Supplementary Figure 3)” by “(see Supplementary Figure 4 for data and further discussion)” (note that Fig. S3 now is Fig. S4).

The reviewer is correct in saying that reduction of *ScAA10C* is slower and this is of course worth mentioning. We have added the following short paragraph at the end of this section in Results:

“Although no reliable rates could be obtained for cellulose-active *ScAA10C*, the data suggested that reduction of this enzyme was slower than reduction of *SmAA10A* (Supplementary Figure 4).”

In addition, we have included the requested additional discussion by adding the following paragraph to the legend of Fig. S4 (previously S3):

“Reliable data could only be obtained for the chitin-active LPMO, *SmAA10A*, for which the difference in fluorescence signal between ground state and reduced copper state is higher, compared to *ScAA10C*. Since the reactions contained spectroscopically active lignin, such a strong signal was needed to obtain reliable data. The stronger signal of *SmAA10A* is likely due to the following: (1) *SmAA10A* has more tryptophans near the copper ion; this improves signal strength; (2) *ScAA10C* has an additional domain that contains tryptophans which give a high “background” fluorescence signal that is less affected by the redox state of the copper. Although no reliable rates could be obtained for cellulose-active *ScAA10C*, comparison of panel c with panels a and b suggests that reduction of this enzyme is slower than reduction of *SmAA10A*.”

Finally, we have, as suggested, added raw data to Fig. S4 (previously S3) to further illustrate the technical issues (new panels d and e).

COMMENT: 4.) In the paragraph starting at line 247, the authors perform their experiments on lignin reduction with dialysed and native lignin in an attempt to remove low molecular weight species from the lignin. Did the authors do anything to check that this had removed some low molecular weight species? I appreciate they state this is normal practice with manganese peroxidases for instance to remove any Mn²⁺ knocking around, but low molecular weight lignin species are likely harder to remove through dialysis so I would expect the authors to have done something to check that there genuinely was an effect on low molecular weight species from this treatment.

RESPONSE: We have not done any further checks beyond what is reported. We agree with the reviewer’s considerations but would respectfully argue that our data for the dark reactions actually do show that we remove something (see Fig. 4). The text (lines 260-263) says: “For reactions in the dark, the dialyzed lignin resulted in lower LPMO activity compared to the already slow reaction with native Kraft lignin (Figure 4a). It is conceivable that under these conditions, the presence of rapidly diffusing low molecular weight reductants has a notable impact on the (low) rate of *in situ* H₂O₂ generation that drives the reaction.”

We could do additional experiments if the Editor or reviewer insist, but we think that such experiments are hardly needed to substantiate the key findings and conclusions of our study.

COMMENT: 5.) In the NMR analysis in Figure 6 and Supplementary figure 6, the figure legends state that the samples containing LPMO had about half the amount of lignin compared to samples without the LPMOs present and so the signals were essentially scaled. I question whether this is a fair thing to do without more significant justification in the main text. The reason for the discrepancies in concentrations should really be explained at the very least. The background noise is clearly higher in the LPMO containing samples so how do the authors know that any similar signals are not simply lost in the noise in the LPMO containing samples as there are clearly some small signals at the same chemical shift in the dark, non-LPMO sample?

RESPONSE: We fully agree that this is an issue that should be addressed more clearly. We also agree that this difference in sample concentration creates the issues raised by the reviewer and it is for this reason we limit the discussion of differences to sets of samples with the same concentration (i.e., light vs dark are compared, and LPMO vs LPMO+Avicel are compared). Any comparisons between these two sets of data are limited to changes above signal-to-noise in the more sensitive proton spectra. The differences in sample concentration are due to differences in sample availability. Importantly, the cinnamaldehyde end group signals seen in the spectrum for light-treated lignin clearly surpass the signal-to-noise ratio in the spectrum for LPMO treated lignin. So the difference is real, but we cannot exclude that we are missing weaker signals in the spectra for the LPMO treated lignin, as the reviewer rightfully points out. To make this clearer to the reader we have made the following changes:

- 1) The difference in concentration and the scaling are more clearly explained in the methods. A new paragraph has been added that explains multiple details of how NMR data were collected and handled (partly in response to comments 3, 5 & 6 by Reviewer #1)
- 2) Figure 6 and Supplementary Figure 7 (previously Supplementary Figure 6) now include panels e) and f) to more clearly highlight the differences, as well as to guide the reader to comparing the correct sets of data. The figure captions have been updated to reflect this. Importantly, the revised captions also make clear that there were no differences visible in the part of the spectrum that is not shown in these new panels. Using Fig. 6 as an example, the following text has been added to the legend:
“Regions of the spectra displaying differences related to treatment with light (f) or an LPMO (e) are shown in the panels to the right. There were no detectable differences in the parts of the spectra that are not shown in panels e and f.”
- 3) The use of two different amounts of lignin is more precisely described in all relevant Figure legends (Fig. 6 & Supplementary Figures 7-10 (previously 6 – 9)).
- 4) In the discussion of Fig. 6 in the Results section, we have added the following sentence:
“It should be noted that the spectra for LPMO-treated lignin have a higher signal-to-noise ratio compared to the spectra for light-treated lignin (due to a 2-fold lower lignin concentration leading to approximately 4-fold lower sensitivity).”
We then further elaborate on this in the Discussion; see next change.
- 5) We have added a cautionary paragraph to the Discussion section explaining the limitations of our NMR work. Among other things, we explicitly acknowledge that we cannot exclude missing certain signals in the NMR spectra for the reaction with LPMO, due to a lower signal-to-noise ratio.

The cautionary paragraph added to the Discussion section reads as follows:

“Importantly, while the structural studies of lignin show effects of both irradiation and LPMO action and clearly point at the chemical processes involved, further studies are needed to fully unravel

structural changes in lignin. We used the highest practical sample concentrations in the NMR analyses, to maximize sensitivity. The complexity and heterogeneity of the lignin structures requires high sensitivity, while achieving complete dissolution of samples is challenging. It is likely that the structural changes in lignin observed in this study only provide part of the picture, due to low signal-to-noise ratios, particularly for the 1D carbon spectra. Of note, the apparent lack of an effect of LPMO treatment on the 1D carbon spectra of lignin (Figure 6 & Supplementary Figure 7) could to some extent be due to the lower signal-to-noise ratio in these spectra (compared to the spectra obtained in the experiments with light). Thus, we cannot fully exclude that LPMO action also leads to lignin oxidations similar to those occurring upon treatment with light. Further in-depth studies of treated and untreated lignin are needed to unravel the full impact of light and LPMO action of lignin. Such studies may eventually allow the determination of quantitative correlations between the degree of lignin oxidation, the amount of hydrogen peroxide produced and LPMO activity. Of note, revealing such correlations would require accurate quantitative detection of all LPMO products and hydrogen peroxide levels under relevant conditions, which is challenging for reactions with lignin.”

COMMENT: 6.) The final results paragraph relating to whether the H₂O₂ could have been generated as a result of H₂O reduction as opposed to O₂ reduction appears out of place to me. This may have been a late addition to the paper or done in response to a previous submission to another journal, but I think it would sit better earlier on following the initial demonstration of H₂O₂ production as a result of light treatment and the demonstration of LPMO activity under such conditions. In addition, none of the MALDI data that are described is shown either in the main text or the SI, this should really be included.

RERESPONSE: We fully agree with the reviewer that the lack of shown experimental data is an issue here and we have done something about this (see below). The placement of this section is deliberate because, in our view, it is a bit of a sidetrack that, nevertheless, must be mentioned since people in the field speculate about these things. We hope it is okay that we did not change its location, and we can assure the reviewer that this was not a “late addition”. Our revision of the text, addressed below, includes changes in the subsection header and the first sentences that likely make the placement more logical.

We have carefully re-assessed our data and concluded that the product analysis by chromatography gives a clear answer to the question raised (-> water oxidation does not or hardly happen and is not the reason for light-driven LPMO-activity). This convincing data is now included (new Supplementary Figure 12). The mass spectrometry data is of relatively low quality, which is due to complications caused by low product levels and the presence of lignin in the samples. Fact is that we do not see any products that would be the result of water oxidation (i.e., oxidation of H₂¹⁸O) and this coincides with the chromatographic data. However, we conclude that we cannot be sure that we would have observed low levels of products resulting from water oxidation if they would have been there. So, from the MS data, we cannot conclude that water oxidation did not happen at all.

We have rewritten both paragraphs of this last subsection in Results, the name of which has been changed from “Probing for light-promoted lignin-driven cellulose solubilization by ScAA10C under anaerobic conditions” to “Probing for a possible role of water oxidation”. We focus on the experimental data that have been added in new Supplementary Figure 12 and we have shortened the discussion of our non-conclusive mass spectrometry data. Since we used ¹⁸O-labeled water and hydrogen peroxide in these experiments, we have kept the discussion of what we wanted to achieve by doing so and why this turned out to be difficult. However, in the revised manuscript, these details now appear as a side note in the legend of Supplementary Figure 12 and not in the main manuscript. Please see the revised manuscript for the revised text.

COMMENT: 7.) Generally speaking, it is hard to see either the industrial or biological relevance of the research as presented. There is no doubt that the results support that light induced H₂O₂ production can occur from lignin (which I believe is a known phenomenon anyway) but how likely is it that this occurs in nature to a significant extent when LPMOs are present? Fungi are the most heavy users of LPMOs but they typically grow in dark, damp places as far as I am aware and so light is unlikely to have significant impact. How do the levels of light used in this research compare to the light intensities that would be expected in the habitats of such fungi or indeed soil dwelling bacteria? I appreciate that the authors note the caveats as relate to the potential industrial applications of this research but I think the likely biological significance of the findings should be discussed in greater depth as well.

RESPONSE: We largely agree with the reviewer. As we write in the paper, the industrial relevance in classical biomass processing may seem limited, although, reactor designs with bypass loops for irradiation have been discussed in the field and is a topic of interest (e.g., see the discussion in Blossom et al., Photobiocatalysis by a Lytic Polysaccharide Monooxygenase Using Intermittent Illumination, ACS Sustainable Chem. Eng. 2020, 8, 25, 9301–9310). We ourselves have noted the impact of light on enzymatic processing lignocellulosic biomass at high dry matter, but systematic studies remain to be done. It is worth noting that lignocellulosic feedstocks that are subjected to enzymatic saccharification typically contain lots of lignin. And, of course, our studies illustrate the potential to drive other (industrial) peroxygenase reactions with light-irradiated lignin, as recently described by Kim et al (Nature Synthesis, 2022, 1, 217–226; we cite this paper).

The biological significance of our findings is (of course) not fully clear. As we outline in the Introduction (Lines 83-92) and on the first paragraph of the Discussion (lines 417-425), it is well known that light promotes biomass turnover. Importantly, such turnover most certainly also happens in the light (e.g., by bacteria) and not only in “dark, damp places”. Of note, H₂O₂ is thought to play multiple roles in fungal biomass turnover (e.g., Fenton chemistry employed by brown rot fungi). All in all, we feel that it’s fully justified to make the link, and to speculate on how the photoreactivity of lignin may play a role in the enzymatic degradation of light-exposed plant material. The many previous observations on the effect of light on conversion on plant biomass deserves a better explanation, and our work provides one. This being said, and while we believe this is important, we do not know how relevant this really is in natural eco-systems, as the reviewer correctly points out.

We feel that it would be wrong to add an extensive discussion (because we do not know), but we have made some adjustments to the manuscript to better address the uncertainty regarding industrial and biological relevance, while at the same time, underpinning the potential of using lignin as a natural light-harvesting material in enzyme catalysis. The changes:

(A) At the end of the fifth paragraph of the Introduction, we have added the following sentence: “Of note, possible effects of light may also be relevant for reactor design in industrial biorefining of lignocellulosic biomass, since pretreated feedstocks that are subjected to enzymatic saccharification with LPMO-containing cellulolytic enzyme cocktails usually contain large amounts of lignin.”

(B) The 2nd half of the 2nd paragraph of the Discussion section has been changed and slightly expanded.

Old text: “It should be noted that the use of light to control LPMO activity in commercial bioreactors operating at high dry matter concentrations with for instance lignocellulose will be challenging as light is heavily attenuated in reaction slurries. It is also worth noting that the present results suggest that the outcome of lignocellulose saccharification experiments with LPMO-containing cellulase cocktails, may depend on the vessel type (glass or steel) and the light conditions in the laboratory or the industrial plant.”

New text: “It should be noted that the use of light to control LPMO activity in commercial bioreactors operating at high dry matter concentrations with for instance lignocellulose will be challenging as light is attenuated in reaction slurries. Still, light will penetrate to some extent and it is thus worth noting that the present results suggest that the outcome of lignocellulose saccharification experiments with LPMO-containing cellulase cocktails may depend on the vessel type (glass or steel) and the light conditions in the laboratory or the industrial plant. These light attenuation issues will not apply in light/lignin fueled reactions with other H₂O₂-dependent enzymes, for example the oxyfunctionalization of hydrocarbons by unspecific peroxygenase recently reported by Kim et al.⁴⁰.”

(C) The 2nd half of the final paragraph of the Discussion section has been changed and slightly expanded.

Old text: “No matter these potential wider implications, the present study provides important insight into the complex roles of lignin and light in biomass conversion and the catalytic potential of LPMOs.”

New text: “While these are interesting possible implications and while the impact of light on biomass conversion in Nature is indisputable, the magnitude and relative importance of light/lignin-fueled catalysis by LPMOs and other H₂O₂-dependent biomass-degrading enzymes remains to be established. No matter the width and magnitude of these implications, the present study provides important insight into the complex roles of lignin and light in Nature and the catalytic potential of LPMOs.”

COMMENT: MINOR CORRECTIONS

Line 89 – explaining should be “explain”

RESPONSE: Corrected

COMMENT: Line 259 – I think it is more normal practice to write “3.7·10³ M.s⁻¹ and 2.9·10³ M.s⁻¹” as “3.7 x 10³ M.s⁻¹ and 2.9 x 10³ M.s⁻¹”

RESPONSE: Corrected

COMMENT: Line 428 – “a cheap and abundant...” should be “a cheap and abundant...”

RESPONSE: “an” has been changed to “and”

COMMENT: Lines 430 to 436 appear to be missing some references.

RESPONSE: Indeed, several more reference should be cited in line 430 (now line 436), and we have now added two: Canella et al., 2016 (already cited) and Dodge et al., 2020 (new reference; Water-soluble chlorophyll-binding proteins from Brassica oleracea allow for stable photobiocatalytic oxidation of cellulose by a lytic polysaccharide monooxygenase. *Biotechnol Biofuels*. 2020 Nov 30;13(1):192.)

We have no references to add at line 436. Here, we make an important point based on the observations described in our paper. We have indeed seen such effects in preliminary work in our own lab, but it is too early to report on that.

COMMENT: Lines 467 to 469 – Is there a reference to the effect of ploughing and how this fits in here?

RESPONSE: This was merely just a thought that we felt was worth mentioning. In response to the reviewer's comment we have rephrased the sentence and added a reference. We modestly believe that this possibility is worth mentioning, but we would be willing to delete this sentence if the Editor or the Reviewer feel that this is a better solution.

Old text: "On another note, our findings suggest that the impact of plowing on biomass conversion in agricultural soils not only relates to access to O₂ but also to access to light that promotes reduction of O₂ to H₂O₂."

New text: "On another note, our findings suggest that changes in access to light may contribute to the well-known impact of tillage regimes on the turnover and sequestration of organic matter in soil⁶⁵."

The new reference is: Mechanisms controlling soil carbon turnover and their potential application for enhancing carbon sequestration; Jastrow, JD, Amonette, JE & Bailey, VL; Climatic Change (2007) 80:5–23; DOI 10.1007/s10584-006-9178-3.

REVIEWERS' COMMENTS

Reviewer #2 (Remarks to the Author):

I have read the revised manuscript and the extensive rebuttal provided by the authors. The ultimate changes to the main text appear to be fairly minimal. For me, most of the comments I made have been addressed reasonably well in the updated manuscript though I think that burying many of the discussions around technical hurdles in the supplementary figures is somewhat unsatisfactory. I sympathise with this as adding such details could disrupt the flow of the main text so overall I think it's acceptable.

The water reduction stuff still feels a bit out of place to me. The inclusion of the data is useful I the revisions have helped but I am not sure that this is truly necessary for this manuscript so I might be tempted to leave it out given the lack of significant insight from the MALDI that appears to have largely not worked in reality.

The main question I am still left with is with regards fit to the journal given the unknowns relating to industrial and biological relevance, I feel this is largely an editorial decision at this stage.

Final revisions for Nature Communications manuscript NCOMMS-22-24127A-Z, entitled:
Visible light-exposed lignin facilitates cellulose solubilization by lytic polysaccharide monooxygenases

Point-by-point response to the reviewers' comments, reproduced verbatim

COMMENT: I have read the revised manuscript and the extensive rebuttal provided by the authors. The ultimate changes to the main text appear to be fairly minimal. For me, most of the comments I made have been addressed reasonably well in the updated manuscript though I think that burying many of the discussions around technical hurdles in the supplementary figures is somewhat unsatisfactory. I sympathise with this as adding such details could disrupt the flow of the main text so overall I think it's acceptable.

RESPONSE: We thank the reviewer for his/her time and are happy that our solutions are found acceptable.

COMMENT: The water reduction stuff still feels a bit out of place to me. The inclusion of the data is useful I the revisions have helped but I am not sure that this is truly necessary for this manuscript so I might be tempted to leave it out given the lack of significant insight from the MALDI that appears to have largely not worked in reality.

RESPONSE: If we understand the reviewer correctly, our revisions, which implied that we simplified this part, were considered satisfactory. Still the reviewer believes that the issue remains a bit out of place, although the reviewer does not seem to be firmly asking to remove the issue. The reviewer points out that the MALDI analysis did not work, to which we agree (and we do not claim anything else in the manuscript).

We believe that the fact that we could not detect LPMO activity in anaerobic conditions using irradiated lignin is relevant and of interest, considering where the field is and considering that (potential) water reduction is a hot and debated topic. This issue is also of industrial importance, since oxygen supply can be a major cost-driver in bioprocessing. We believe that, after reading the preceding part of the paper, quite some readers will ponder about the role of oxygen and the possibility of water reduction. Except from the MALDI data (see next paragraph) the results of the experiments are clear, conclusive, and undisputed. Therefore, all in all, we would like to keep this short section at the end of the Results section.

The section ends with the following text: "We did these experiments in $H_2^{18}O$ and used $H_2^{18}O_2$ in the control reaction with hydrogen peroxide, because such an approach in principle could provide additional evidence for (the absence of) water oxidation, as explained in the legend of Supplementary Figure 12. Unfortunately, due to the presence of lignin, the quality of MALDI-TOF MS spectra was too low to provide additional support for the conclusions drawn from chromatographic product analysis." We are totally willing to remove this text (and the corresponding explanatory text added to the legend of Fig. S12; now Fig. S13). However, since doing this type of experiments is the "logical thing to do", we think it is very useful for readers to know that they did not work (i.e., a technical limitation) due to the presence of lignin.

COMMENT: The main question I am still left with is with regards fit to the journal given the unknowns relating to industrial and biological relevance, I feel this is largely an editorial decision at this stage.

RESPONSE: No further author response required, we believe.